# Evidence accumulation in the pre-supplementary motor area and insula drives confidence and changes of mind

Dorian Goueytes [1,2] ✉, François Stockart [1], Alexis Robin [3,4], Lucien Gyger [1], Martin Rouy[1], Dominique Hoffmann[5], Lorella Minotti[3,4], Philippe Kahane [3,4], Michael Pereira [3,6] & Nathan Faivre [1,6]

Evidence accumulation is a powerful mechanism to explain the temporal dynamics of decisions, as well as their metacognitive components such as confidence judgments and changes of mind. However, it is still unclear how and where in the brain evidence accumulation leads to these two metacognitive components. We report intracranial high-gamma activity in patients with epilepsy recorded while they perform a visual discrimination task and estimate their confidence level. Our results indicate an anatomical overlap between the neural correlates of evidence accumulation, confidence, and changes of mind in the pre-supplementary motor area, as well as in the orbitofrontal, inferior frontal, and insular cortices. Behavioural and electrophysiological results are reproduced with a post-decisional evidence accumulation model, and the temporal dynamics of decision-making is characterized with mouse-tracking and intracranial electrophysiology. We conclude that confidence and changes of mind result from evidence accumulation, instantiated before the decision in the pre-supplementary motor area, and after the decision in the insula.

Most of our decisions are accompanied by a feeling of confidence. This feeling, which is thought to derive from metacognitive processes[1], is essential in a world where immediate feedback rarely follows decisions, and where agents must quickly adjust or even change their decisions based on internal evaluations[2]. In recent years, evidence accumulation models have been successfully used to describe how decisions unfold over time[3–5], and to understand confidence judgements and changes of mind (CoMs) as part of the decision-making process[6–9]. According to *bounded evidence accumulation*, sensory evidence is accumulated until a decision bound is reached[10,11]. Some computational and neurophysiological evidence suggests that the state of the accumulators at the decision time can explain confidence[12,13]. However, there is also compelling evidence that both confidence[7,14–16] and CoMs[9,17] are well described by models assuming

that evidence continues to accumulate after the decision bound is reached (i.e. post-decisional evidence accumulation, see in ref. [18]). Overall, the nature of the transformation between accumulated evidence and confidence remains debated[7,16,19]. Moreover, its neural implementation is also unclear, with numerous discrepancies depending on the model, task, and imaging methods [13,15,20–25]. To characterise how decisional and post-decisional processes unfold in time, both at the behavioural and neural levels, we combined stereotactic electroencephalography (sEEG) and computational modelling to explore the neural implementation of evidence accumulation, confidence judgements, and CoMs.

We used a protocol to precisely track the decisional and post-decisional temporal dynamics of patients with epilepsy implanted with sEEG electrodes as they made perceptual decisions. Patients used a

[1]Université Grenoble Alpes, Université Savoie Mont Blanc, CNRS LPNC, Grenoble, France. [2]SCALab, Université Lille, UMR 9193, Villeneuve d'Ascq, France. [3]Université Grenoble Alpes, Inserm, U1216, Grenoble Institut Neurosciences GIN, Grenoble, France. [4]Department of Neurology, Grenoble Hospital, Grenoble, France. [5]Department of Neurosurgery, Grenoble Hospital, Grenoble, France. [6]These authors jointly supervised this work: Michael Pereira, Nathan Faivre. ✉e-mail: dorian.goueytes@univ-lille.fr

computer mouse to respond to a two-alternative forced-choice visual discrimination task, followed by a confidence rating task. Kinematic analyses of mouse trajectories allowed us to infer both decision times (i.e. when patients started to move the mouse to respond) and CoMs (i.e. when patients modified their trajectory to respond), which we used to temporally realign sEEG data. We harnessed the broad coverage and fine spatial resolution of sEEG to isolate neural markers of evidence accumulation and explored how these markers overlap with markers of confidence and CoMs. Based on an evidence accumulation model, we expected to find early neural correlates of evidence accumulation to determine decision time and accuracy, as well as later correlates to continue after the decision to guide confidence and CoMs. Together, our results support the co-existence of pre- and post-decisional evidence accumulation in the cortex and disentangle their relative contribution to decisions, confidence, and changes of mind.

## Results

Twenty-four patients with drug-resistant focal epilepsy performed a two-alternative forced-choice visual discrimination task. They had to indicate whether a random-dot kinetogram was moving toward the right or left side of the screen. Patients reported their choice by moving a cursor with the mouse to click on one of two circles in the top-left and top-right corners of the screen, corresponding to each motion direction (Fig. 1A). Mouse-tracking allowed us to define movement onset as a proxy for decision time and to detect CoMs and their onsets through changes of trajectories. This visual discrimination task was followed by a confidence judgement where patients were asked to rate their confidence in their initial choice on a continuous scale from 0 to 100 (0: Sure incorrect, 50: Unsure, 100: Sure correct).

### Behavioural results

All patients were able to perform the task, with an average response accuracy computed at click time of 78.76% (sd = 6.78%), an average confidence judgement of 78.81% (sd = 15.27%), a mean decision time of 1.43 s (sd = 1.1 s) and a mean response click time of 2.89 s (sd = 1.36 s). All trials with decision time >2.5 s (17.02% of total trials) were excluded from further analyses, as they were likely to correspond to lapses in attention and/or distraction due to the hospital setting and could

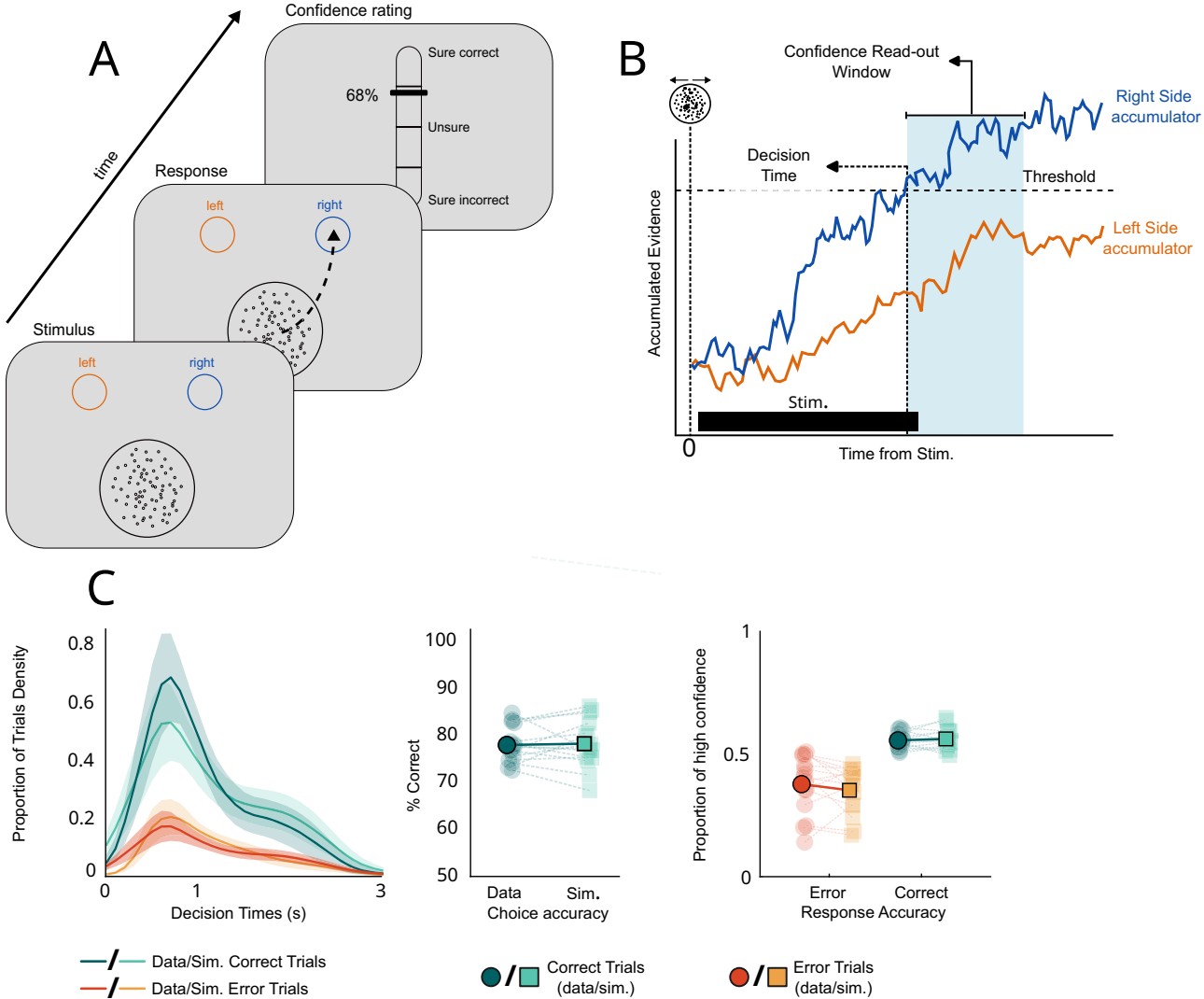

**Fig. 1 | behavioural and modelling results. A** Behavioural paradigm. Patients observed the random dot kinetogram and reported its motion direction using the mouse while it remained on screen. Confidence was rated on a continuous scale ranging from 0 to 100. **B** Schematic description of the race diffusion model used to fit data. A decision was defined when one of two accumulators coding for each possible motion direction crossed a threshold. Confidence was computed as a read-out of evidence accumulated up to 1 s after decision time (confidence read-out window, in blue). **C** Comparison of simulated and data performance. The left panel shows the observed and modelled decision time for correct (blue) and error (orange) trials (full lines represent mean density and shaded areas represent 95% CI). The middle panel shows the observed and modelled response accuracies. The right panel shows the observed and modelled proportions of trials with low and high confidence for correct (blue) and error (orange) trials. Individual averages are plotted in light blue/orange and population averages in dark blue/orange (n = 15).

confound the analysis of sEEG data. Note that all neural analyses were also performed without excluding these trials, which produced qualitatively similar results (see SI–Supplementary Note, Fig. S8–9). Using generalised linear mixed-effects regressions, we found an effect of response accuracy on decision times ($\beta = -0.2$, CI = ±0,18, $p < 0.001$) and on confidence ($\beta = 1.84$, CI = ±0.8, $p < 0.001$), indicating that patients were quicker and more confident in correct vs. incorrect trials. The relationship between confidence and accuracy at decision time was confirmed by computing the area under the receiving operating characteristic curve (AUROC = 0.59, SE = 0.02). Results obtained in individuals with epilepsy were compared with those obtained in healthy controls (see SI - Supplementary Note, for details). We found no significant difference between patients and controls in terms of performance (Mann–Whitney U test, U-stat = 125, $p = 0.60$), decision times for correct (Mann-Whitney U test, U-stat = 123, $p = 0.66$) and error trials (Mann-Whitney U test, U-stat = 115, $p = 0.91$), and proportion of low, mid or high confidence judgements for correct and error trials (Mann–Whitney U test with Benjamini-Hochberg false discovery rate correction, respective U-stats: low correct/error = 85.5/97, mid correct/error = 58/112, high correct/error = 155/111 $p > 0.01$).

### Response accuracy, decision times, and confidence explained by accumulated evidence

To assess whether evidence accumulation is a possible mechanism underlying our behavioural data, we fitted an evidence accumulation model consisting of two anti-correlated accumulators (one for each possible motion direction) to decision times and response accuracy[16,26]. We simulated confidence judgements as a readout of accumulated evidence between 0 and 1 s following the initial choice (Fig. 1B; see Methods). The model reproduced both decisions (Figs. 1C and S1A, B) and confidence (Figs. 1C and S1C) data well (all R (Spearman) > 0.6, $p < 0.017$). We further showed that in 13/15 of the patients, models considering post-decisional evidence accumulation had higher log-likelihoods than models considering evidence accumulation until decision time (i.e. accumulation-to-bound). The average fitted confidence readout time was 0.30 s (SE = 0.09), and longer confidence readout times were associated with higher AUROCs in both simulated ($R = 0.79$, $p < 0.001$) and observed data ($R = 0.73$, $p = 0.0021$), indicating that post-decisional evidence accumulation contributes to accurate confidence judgements (see also Fig. S10). We also fitted the post-decisional evidence accumulation model to our control group data with similar results (Fig. S2). Although we note that other variations in the way confidence is computed have been proposed[16,18,27], our results indicate that evidence accumulation is a plausible mechanism underlying perceptual decision-making and confidence judgements in this task.

### Identification of task-selective sEEG channels

We recorded from a total of 1138 channels (each channel corresponding to the bipolar referencing of two physical contacts in the brain). We selected 414 channels that were localised in predefined regions of interest (ROIs) corresponding to areas susceptible to instantiate evidence accumulation or to play a role in confidence judgements based on the literature[15,20,28–31]. Namely, we selected channels in the visual (caudal medial visual cortex, rostral medial visual cortex, cuneus), parietal (superior parietal cortex, medial superior parietal cortex, dorsal inferior parietal cortex), dorso-lateral prefrontal (dlPFC), inferior frontal (IFC), orbitofrontal (OFC), and insular cortices, as well as the pre-supplementary motor area (pSMA; see SI - Supplementary Note for corresponding Brodmann areas). We restricted our analyses to stimulus-responsive channels, defined as showing a significant difference between a baseline window (−500ms to −100 ms relative to stimulus onset) and a post-stimulus window (200 ms up to median decision time) (Mann–Whitney U test, critical $p$ value 0.05). A total of 102 channels were selected, and all subsequent analyses were

performed on this subset (see SI - Supplementary Note for results from non-selected channels). We focused our analyses on the high-gamma frequency band (70–150 Hz), a proxy for local field potentials[32]. Next, we compared high-gamma activity in correct and incorrect trials. We found 6 channels out of 102 within our ROIs with higher activity following correct vs. incorrect decisions (permutation test, $p = 0.030$, 1000 permutations), and none with increased high-gamma activity for incorrect trials. Because of this low number of identified channels related to error processing, we focused the rest of the analysis on correct trials (see SI - Supplementary Note for analyses including incorrect trials, providing similar results, Fig. S8–9).

### Identification of sEEG channels reflecting evidence accumulation

We aimed to identify channels exhibiting hallmarks of evidence accumulation[10,15,33–36]. We looked for channels that exhibited a ramping-up of high-gamma activity following stimulus onset, identified by a significant negative correlation between the slope of this ramping-up and decision times across individual trials. To do so, we fitted a linear function to the high-gamma activity for every correct trial from stimulus onset up to the decision time. Then, we selected channels for which slopes were significantly negatively correlated with decision times (Pearson R test, critical $p$ value 0.05), and found that 41 channels out of 102 exhibited this correlation (thereafter referred to as EA-candidate channels). We assessed the significance of this result by repeating the analysis while randomly permutating the decision times with regard to high-gamma activity. This strategy ruled out that the observed slopes stemmed from the inverse correlation between the slope and decision time built into the analysis (i.e. steeper slopes may be found within shorter epochs among trials with short decision times). We obtained significant results (1000 permutations, $p = 0.001$, see methods), indicating that our effect was indeed linked to a ramping-up of high-gamma activity and not a statistical artefact. Interestingly, channels reflecting evidence accumulation were found across all selected regions of interest (Fig. 2 and SI - Supplementary Note Table 2). The widespread repartition of EA-candidate channels corroborates recent results using whole-brain fMRI and intracranial recordings, showing that accumulators are spatially distributed across the brain[33,36,37].

### Identification of sEEG channels reflecting confidence

Next, we examined the relationship between single-channel activity before correct decisions and confidence. To do so, we correlated mean high-gamma activity in a 400 ms window preceding decision time with confidence judgements. We found 9 such channels in the visual cortex, parietal cortex, and the pSMA, corresponding to 21.95% of the EA-candidates channels (permutation test: $p = 0.0019$). To support the results of our model indicating that confidence was better predicted when allowing evidence accumulation to continue after the decision, we then aimed to establish if the activity of EA-candidate channels correlated with confidence following decision time. To do so, we correlated mean high-gamma activity in a 400 ms window following decision time with confidence judgements. We found 11 channels with a positive correlation (Spearman Rank Order correlation test), corresponding to 26.82% of the EA-candidate channel pool (permutation test: $p = 0.001$, SI - Supplementary Note Table 2) (Fig. 3). Five of these channels (all but one localised in the pSMA and the OFC) also reflected pre-decisional evidence accumulation. Interestingly, the subset of channels reflecting post-decisional confidence was primarily localised in frontal areas (81.81 % of channels, 9 out of 11, 7 in the insula and 2 in the OFC, with the two remaining channels located in the visual cortex). This result shows that a significant proportion of channels exhibiting evidence accumulation-like activity following stimulus onset also co-varied with confidence after the decision, supporting the view that post-decisional evidence accumulation subserves confidence

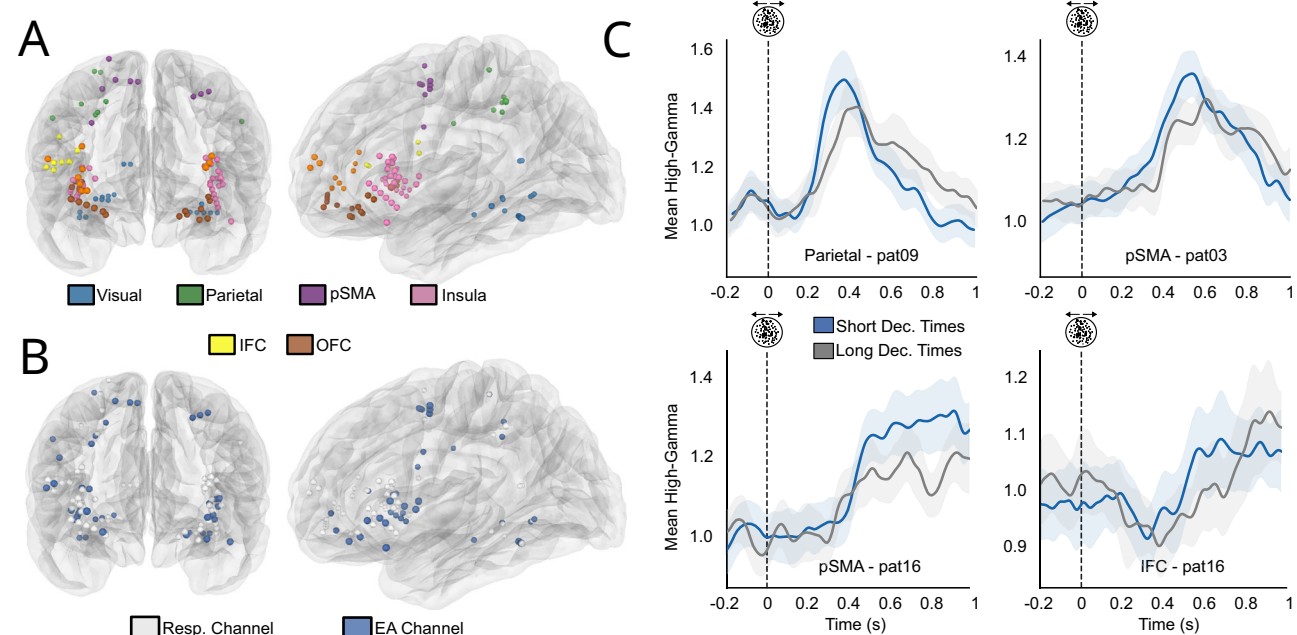

**Fig. 2 | EA-candidate channels in correct trials. A** Channels from all patients are displayed on a template brain based on MNI coordinates. Responsive channels are shown in colours according to their ROIs. **B** Responsive channels reflecting evidence accumulation (blue) or not (white) displayed on the same glass brain as (**A**). **C** Examples of EA-candidate channels. Each graph represents trial-averaged high-gamma activity (in arbitrary units) aligned on stimulus onset for all correct trials for a single channel. Even if statistics were performed on continuous decision times, and for illustration purposes only, blue lines correspond to averaged high-gamma for the 50% fastest trials and light grey lines the 50% slowest. Shaded areas represent 95% CI.

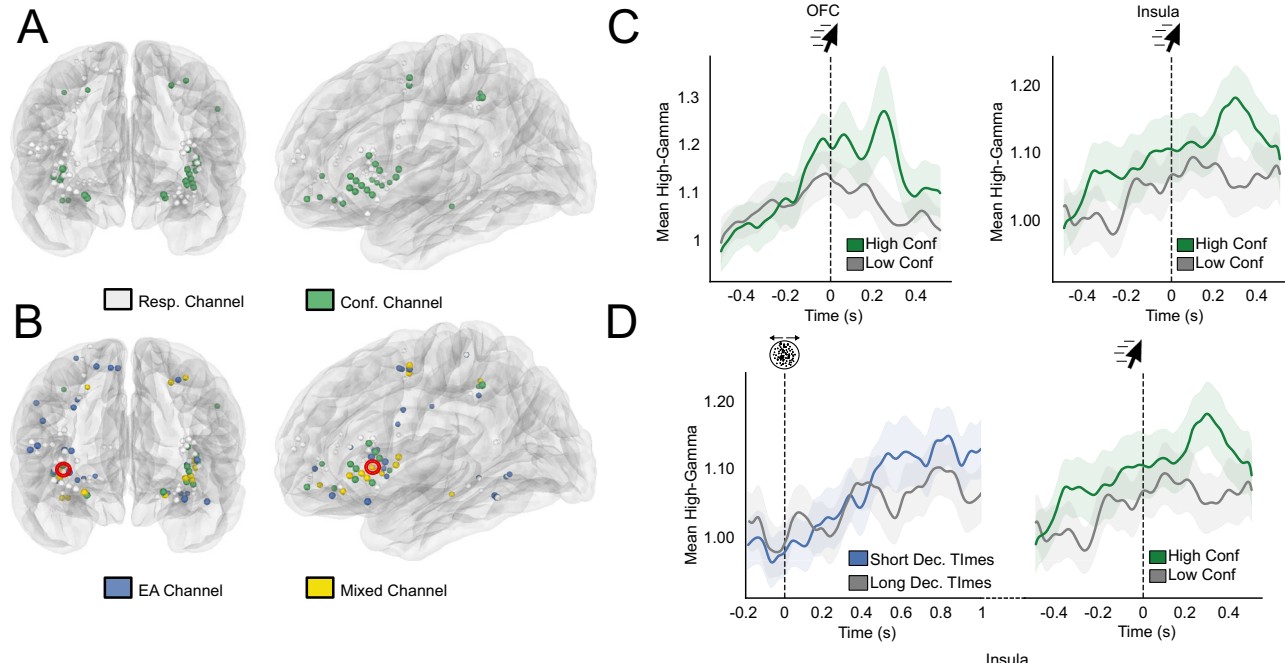

**Fig. 3 | Repartition of EA-candidate channels and channels reflecting confidence in correct trials. A** Confidence-sensitive channels plotted in green against all responsive channels in white (**B**) representation of responsive channel (white), EA-candidate channels (blue), confidence channels (green) and channels with an overlap between evidence accumulation-like activity and confidence (yellow). **C** Example of two confidence-sensitive channels. Trial-averaged high-gamma activity (in arbitrary units) is plotted aligned on decision time. Green lines correspond to the 50% of trials with the highest confidence and grey to the 50% with the lowest confidence. Shaded areas correspond to 95% CI. **D** Example of a channel (highlighted in red in (**B**)) with an overlap between evidence accumulation-like activity and confidence. The left panel corresponds to stimulus-aligned, trial-averaged high-gamma separated based on decision time. The right panel corresponds to movement-aligned high-gamma activity separated based on confidence. Shaded areas correspond to 95% CI.

**Table 1 | Summary of ROI analyses**

| | Correlates of Evidence-accumulation | | | Correlates of Confidence | | | CoM |
|---|---|---|---|---|---|---|---|
| | Single-channel analysis | ROI analysis | | Single-channel analysis | ROI analysis | | ROI analysis |
| | | EA | Post-decisional effect | | Confidence | Post-decisional effect | |
| Visual Cortex | 8 Channels | No | No | 2 Channels | No | No | No |
| Parietal Cortex | 4 Channels | No | No | 3 Channel | Yes | No | No |
| pSMA | 7 Channels | Yes | Yes | 2 Channel | Yes | No | Yes |
| dlPFC | 1 Channel | No | No | 1 Channel | No | No | Yes |
| IFC | 6 Channels | Yes | Yes | 0 Channel | No | No | No |
| OFC | 4 Channels | No | No | 6 Channels | Yes | Yes | Yes |
| Insula | 11 Channels | Yes | Yes | 14 Channels | Yes | Yes | Yes |

judgements. We also performed a similar analysis, including all 102 responsive channels, and found that 10 additional channels reflected confidence preceding the decision, and 5 following the decision, which were not labelled as EA-candidate channels (permutation test on all responsive channels, $p = 0.001$, see methods). All of these channels were also located in the OFC (2 channels) and the insula (3 channels; see Fig. 3 and SI - Supplementary Note). As decision time and confidence are correlated[38,39], we verified that the magnitude of the correlation between high-gamma activity and decision time was significantly lower than that of the correlation between high-gamma activity and confidence (see SI - Supplementary Note).

### Analysis of regions of interest
To substantiate the single-channel analyses and characterise neural activity at the level of cortical regions, we pooled all EA-candidate channels across all patients for each ROI. Assuming that high-gamma activity related to evidence accumulation ramps up steeper for earlier decision times, we expected to find a relationship between high gamma and decision time after stimulus onset and before decision time[40]. We performed generalised linear mixed-effects regressions between high-gamma activity and decision times at each timepoint following the stimulus onset (while correcting for multiple comparisons, see Methods) to test this hypothesis. Interestingly, this analysis at the ROI level brushes a more nuanced picture of the anatomical distribution of evidence accumulation-like activity, as visual (8 EA-candidate channels), parietal (4 EA-candidate channels), and orbitofrontal (4 EA-candidate channels) areas did not exhibit a significant effect at the ROI level (see Table 1). Only the pSMA (7 EA-candidate channels), the IFC (6 EA-candidate channels), and insula (11 EA-candidate channels) qualified as compatible with evidence accumulation ROIs (Fig. 4). Examining the time course of regression coefficients between high-gamma activity and decision times revealed that the insula and the pSMA had similar peak effect latencies after the stimulus onset (pSMA: 537 ms, insula: 553 ms; Fig. 4D). However, significant regression coefficients were maximal before the decision in the pSMA, while they were observed after the decision in the insula. These results suggest that evidence accumulation might take various forms, with different functional roles and temporal profiles, as shown previously using fMRI[37,41].

We used a similar approach to explore the relationship between high-gamma activity and confidence at the ROI level. We assessed if the pooled high-gamma activity of EA-candidate channels in each ROI also covaried with confidence after the decision time, similar to what we observed with single channels. Among ROIs reflecting confidence (see Table 1), two different temporal profiles emerged. The first group of areas, including the parietal cortex and the pSMA exhibited early peak effect latencies (average peak latency −246 ms) with short-lived effects disappearing before decision time (Fig. 5A, SI−Supplementary Note Fig. 3A). By contrast, the orbitofrontal and insular cortices showed later peak effect latencies (average −134 ms after decision time), with

long-lasting effects extending well beyond decision time (Fig. 5B, C, SI−Supplementary Note Fig. 3B). Although only the pSMA and insula exhibited both markers of evidence accumulation-like activity and confidence at the ROI level, a majority of regions with EA-candidate channels also correlated with confidence. To ensure that these effects were not due to a contribution of decision times, we ran a similar mixed-effects regression including confidence, decision time, and their interaction (see SI−Supplementary Note). This model yielded similar results as the ones described above, with minimal interaction effects. This indicates that the effects of confidence on high-gamma activity reported above are not driven by decision time. Finally, to assess if our computational model could account for these temporal dynamics, we simulated traces of evidence accumulation using the individual parameters found to best reproduce our behavioural data (see above). By manipulating the window during which evidence accumulation could take place following decision time, we checked if the different profiles observed in our neural data could be the results of distinct pre or post-decisional evidence accumulation processes (see methods). Limiting the duration of post-decisional evidence accumulation to the decision time, effectively limiting the model to pre-decisional evidence accumulation, we could reproduce early/transient correlates of confidence similar to the ones we found in the pSMA. By contrast, allowing post-decisional accumulation to take place until the end of the trial resulted in traces resembling the late and sustained correlates of confidence we found in the insula (Fig. 5A, B bottom panels; Fig. S6).

### Behavioural and neural markers of changes of mind
Having established a link between confidence and post-decisional evidence accumulation, we used a similar approach to explain another manifestation of metacognitive monitoring: CoMs. Indeed, CoMs have been proposed to depend on confidence[16] or post-decisional evidence accumulation[9,16,42]. In the context of our experiment, a CoM was defined as a sudden inflexion in mouse trajectory leading to the crossing of the midline between the two targets and to a response corresponding to the side opposite to the one predicted by the initial movement (Resulaj et al.[9], see Methods). We found that the initial movement direction was predictive of the final response in 90.36% of trials, indicating that patients started moving the mouse only when they had made a decision, as per our instructions. The remaining 9.64% of trials (sd = 5.88%, Fig. 6A) were considered CoMs (we verified that initial movements in those trials were indeed directed towards a target, and did not reflect a non-directional movement). Using the same post-decisional evidence accumulation model as for confidence, we simulated the onset of CoMs as the time following the decision when the difference between the two accumulators reached a threshold. By fitting this threshold to the behavioural data of individual patients, our model could predict the distribution of CoM onset timing (Fig. 6B, SI−Supplementary Note, Fig. S1D). Turning to sEEG data, we compared high-gamma activity aligned on CoM onset to the activity of EA-

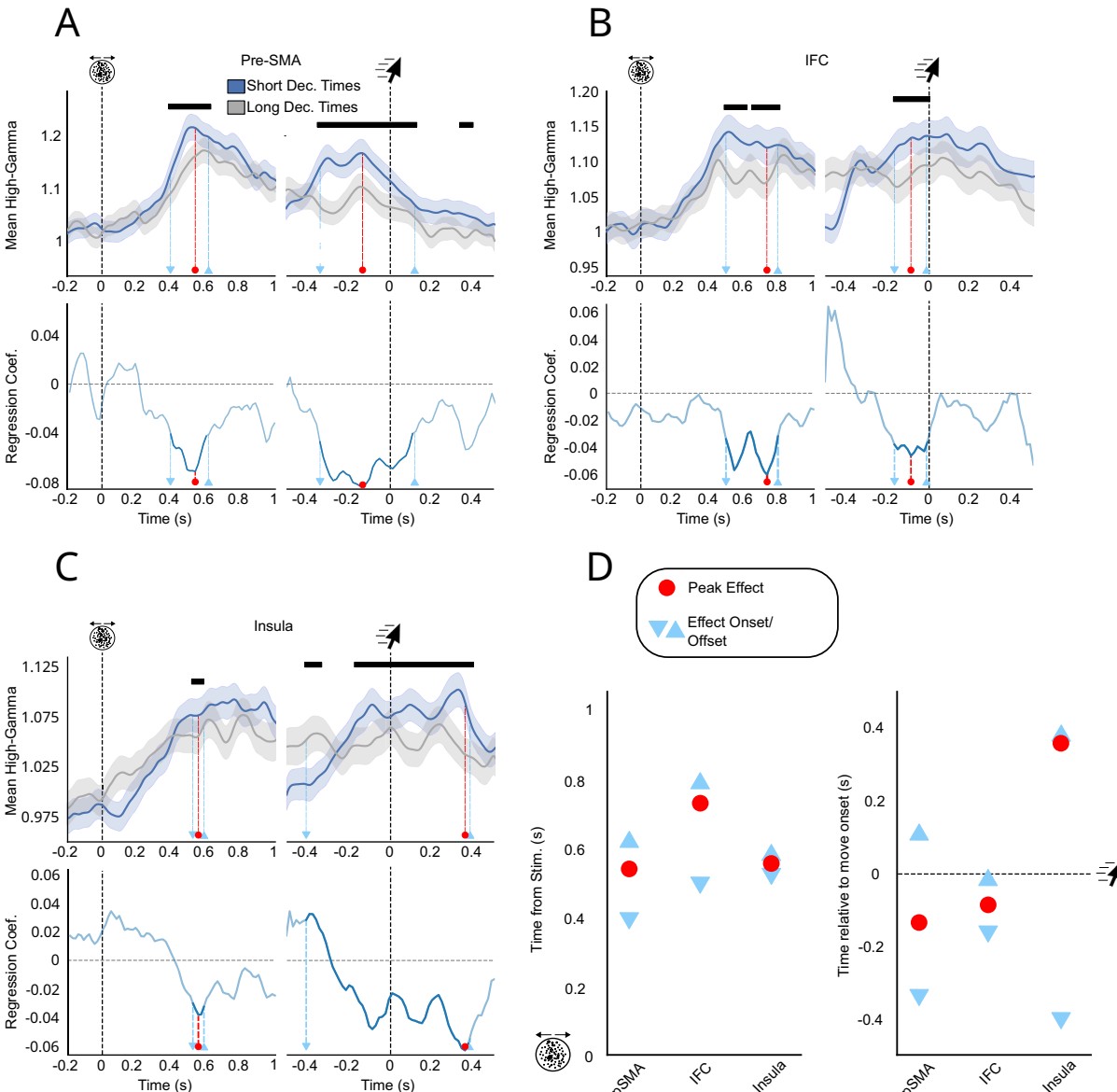

**Fig. 4 | Regional Correlates of Evidence Accumulation.** Stimulus-aligned and movement-aligned high-gamma activity for regions of interest compatible with evidence accumulation **A**–**C** Top: stimulus-aligned and movement-aligned high-gamma activity (in arbitrary units) averaged across patients for all EA-candidate channels. Dark bars indicate a significant relationship between high-gamma and move onset timing ($p < 0.05$, FDR-corrected). Blue lines correspond to averaged high-gamma for the 50% fastest trials, and grey lines for the 50% slowest. Shaded areas correspond to 95% CI. All analyses were performed on continuous decision times and binarized only for illustrative purposes. Bottom: regression coefficient as a function of time in the stimulus- and movement-aligned window. Light blue markers indicate the onset and offset times of significant effects, and the red marker indicates peak effects. **D** Latency analysis in the stimulus and move-aligned window for the three ROIs compatible with evidence accumulation. Conventions are similar to (**A, C**).

candidate channels aligned on the initial decision time. We reasoned that if CoMs involve evidence accumulation, high-gamma activity aligned on CoM onset should closely resemble the activity observed around the initial decision time in channels exhibiting evidence accumulation-like activity. Thus, we defined an ROI as reflecting CoM if its activity was (1) ramping up in the interval leading to CoM and different from similarly aligned activity in non-CoM trials (CoM-like time) and (2) showed a similar pattern of high-gamma activity leading to CoM in CoM trials and to move onset in non-CoM trials. Such ROIs included the parietal cortex, pSMA, OFC, and insula (Fig.6 and Tables 1, and S2). Interestingly, these overlapped both with ROIs which activity was compatible with evidence accumulation and ROIs reflecting confidence. We note, however, that the same analysis restricted to EA-candidate channels did not yield any significant result, possibly due to lack of statistical power, as CoMs were rare.

## Discussion

We studied how evidence accumulation underlies confidence and CoMs during perceptual decision-making. To this end, we fitted a computational model of competing accumulators to patients with epilepsy's decisions, response times, and confidence ratings. Examining how intracranial recordings reflected these behavioural variables, we found overlapping regional markers of evidence accumulation, confidence, and CoMs in the pSMA and the insula.

### Evidence accumulation and confidence

Activity in the pSMA reflected confidence following stimulus onset until the decision time, in line with a possible role of the pSMA in tracking the uncertainty of a task as it unfolds until a decision is reached[29,43–46]. This mechanism may be useful in situations where one needs to adapt behaviour before a decision, for instance when

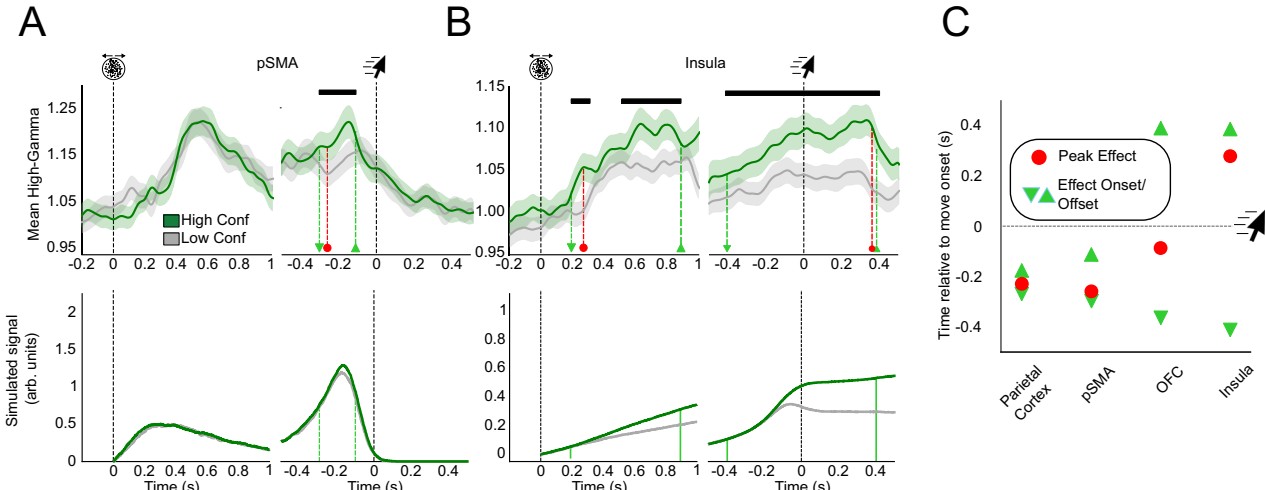

**Fig. 5 | Stimulus-aligned and movement-aligned high-gamma activity in the pSMA and insula as a function of confidence. A, B** Top: stimulus-aligned and movement-aligned high-gamma activity (in arbitrary units) averaged across patients, including only EA-candidate channels in each ROI. Dark bars indicate a significant relationship between high-gamma and confidence ($p < 0.05$, FDR corrected using the Benjamini-Hochberg method). Green lines correspond to averaged high-gamma for the 50% of trials with the highest confidence, and grey lines for the 50% lowest. Shaded areas correspond to 95% CI. Dashed light green markers indicate the onset and offset times of significant effects, and dashed red markers indicate peak effects. All analyses were performed on continuous confidence ratings and binarized only for illustrative purposes. Bottom: average difference between accumulator traces simulated by the evidence accumulation model, when it stopped around the decision time (**A**), or continued after the decision until the end of the trial **B**, **C** Latency analysis in the movement-aligned window for the four regions sensitive to confidence. Conventions are similar to **D** in Fig. 4.

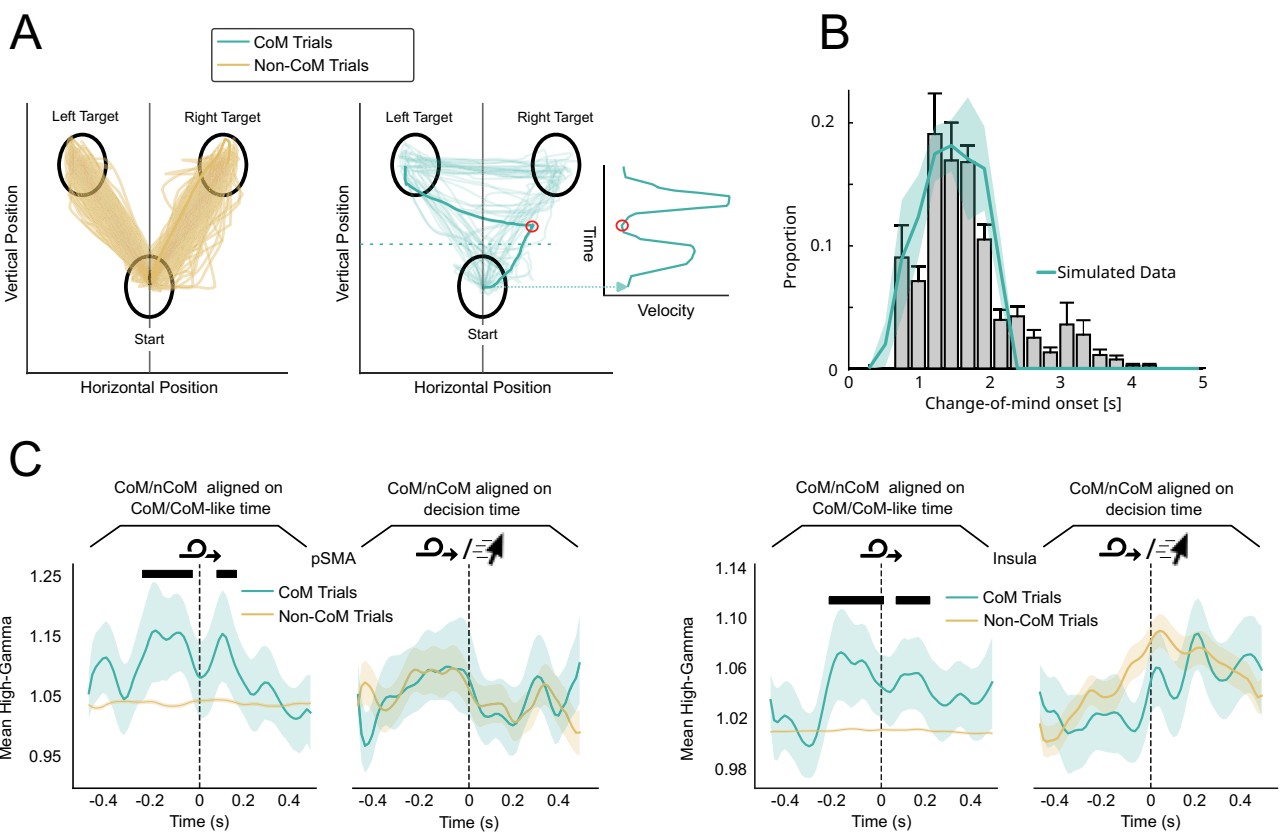

**Fig. 6 | Neural correlates of CoMs. A** Mouse trajectories and velocity profile for regular trials (left) and CoM trials (right) for one example participant. The red circle indicates the onset of a CoM, defined by a change in the mouse direction and velocity. **B** Comparison of observed and simulated CoM onset. Blue line shows simulated data. Grey histogram corresponds to observed CoM-onsets binned and averaged across subjects ($n = 15$). **C** CoM-aligned high-gamma activity for CoM trials compared to high-gamma activity (in arbitrary units) aligned on CoM-like times for non-CoM trials (left panel) or move-aligned high-gamma for non-CoM trials (right panel). Dark bars indicate a significant effect of condition (CoM/non-CoM) on high-gamma levels ($p < 0.05$, FDR corrected using the Benjamini-Hochberg method). Error bars and shaded areas correspond to 95% CI.

deciding to opt out before the decision is taken if it is deemed too difficult[13,47]. The dynamic we observed in the pSMA is consistent with evidence accumulation stopping in the vicinity of decision time[5,11]. Within such a framework, confidence is derived from the state of the losing accumulator, as by definition, the state of the winning accumulator is equal to the bound[13]. The higher the state of the losing accumulator (i.e. the closer it is to the bound), the lower the confidence. At the behavioural level, however, confidence and metacognitive sensitivity of confidence ratings across patients (i.e. how confidence tracked task-performance) were better accounted for by post-decisional evidence accumulation. Interestingly, our model could reproduce the pre-decisional confidence-related activity under the assumption that the signal observed was related to pre-decisional evidence accumulation, but that trials were sorted according to confidence computed post-decisionally (and thus elsewhere in the brain). Therefore, although the pre-decisional correlates of confidence we found could be directly related to pre-decisional confidence computations, they might also arise from a post-hoc sorting of trials based on post-decisional confidence, supported by regions such as the insula (see below).

As the state of each of the two accumulators could not be distinguished at the sEEG level (i.e. we could not find univariate or multivariate signatures of rightward vs leftward visual motion), it is unclear what feature of evidence accumulation was reflected by high gamma activity observed in the pSMA. We considered the *balance-of-evidence* hypothesis[48] stating that confidence is proportional to the difference between accumulators[12,16,49]. Simulating how this difference between accumulators evolves over time, we could qualitatively reproduce the observed high-gamma activity (see Fig. S11 for alternatives). We opted for a readout occurring at a fixed latency after the decision (with some variability in the non-decision time), similar to previous works[7,16,31]. We note that other studies have hypothesised that the readout occurs when accumulated evidence reaches additional collapsing bounds[19,50], without significantly changing the signatures[7,51] of the so-computed confidence.

Interestingly, we could not confirm that activity in the pSMA increased after errors (Fig. S5), despite numerous studies suggesting a role of the pSMA in error monitoring[44,52,53]. One possible reason is that our task included no response conflict and produced long response times, contrary to what is commonly used in experiments showing increased activity in the pSMA following errors[44,52,54]. Our computational model adds support to this account as it simulated lower activity for errors. According to this account, incorrect trials are similar to low-confidence correct trials with small differences between the losing and winning accumulators. Linking cortical activity to precise features of the evidence accumulation process will require more research, including cortical recordings of many individual neurons.

## Post-decisional evidence accumulation in the insula

We also found positive correlations between confidence and high-gamma activity in the insula. In contrast to our findings in the pSMA, correlates of confidence in the insula extended until after the decision. Our computational model fitted on behavioural data could reproduce qualitatively similar traces when evidence accumulation was allowed to continue after the decision and after the estimated confidence readout time. Since the same channels in the insula also exhibited hallmarks of evidence accumulation, our simulations suggest that post-decisional evidence accumulation is a plausible mechanism to explain our data. The duration of the effect of confidence suggests that this post-decisional evidence accumulation is sustained beyond the confidence readout and could support later CoMs. The anatomical dissociation between pre- and post-decisional evidence accumulation was further supported by univariate analyses: EA-candidate channels also encoded pre-decisional confidence in the pSMA and post-decisional confidence in the insula. Interestingly, in the context of

decision-making, stimulating the insula has been found to have an effect on confidence[55]. The insula has also been hypothesised to provide inputs to dorsomedial regions, including the pSMA[56], as well as to the OFC[57,58], another region found to reflect confidence in our data. In addition, contrary to what we found in the pSMA, high-gamma activity in the insula and the OFC was higher for correct than incorrect trials, compatible with the view that these regions are also involved in error monitoring and valuation of a decision[59] (Fig. S6). An alternative interpretation of the insula's post-decisional contribution to confidence judgement would be that a first insular population of neurons implements evidence accumulation following stimulus onset, and then a second insular population, disjoint from the first, implements the estimation of confidence after the decision. Future single-neuron recordings will be needed to confirm that the same population of insular neurons is involved in evidence accumulation and post-decisional confidence.

## Evidence accumulation and changes of mind

Post-decisional evidence accumulation was particularly relevant in our task, as perceptual information was still present on screen following the initial decision, and patients had some time to change their mind. In other words, even after patients initiated a mouse movement towards one of the two targets, they could consider additional sensory evidence and revise their initial decision to reorient the mouse towards the other target, which they did in about 10% of trials[9,16]. It should be noted that the CoMs in our study were observed despite the absence of time pressure to respond. In this sense, CoMs do not reflect a situation where patients hastily moved the mouse to one side or the other to comply with the instruction, and then made an actual decision in a second step. On the contrary, the CoMs we observed were likely endogenous, so that the first observed mouse movement reflected a genuine decision rather than a random movement. The lack of time pressure increases the ecological validity, but results in fewer CoMs. Overall, the effect of CoMs on accuracy and confidence was similar to what had already been reported using the same task in a cohort of healthy participants and participants with schizophrenia[26], see SI− Supplementary Note. Interestingly, we could predict the timing at which such CoMs occurred using our computational model when it allowed evidence accumulation to continue after the decision time. In line with this modelling result, we found that neural activity in the insula−which reflected post-decisional evidence accumulation and confidence−also reflected the occurrence of a CoM. Namely, the neural activity time-locked to the onset of CoMs resembled the activity time-locked to decision-time in trials without CoMs, suggesting that patients changed their minds by initiating a new decision following post-decisional evidence accumulation. This hypothesis is in line with earlier works indicating that continued processing of information following the initial decision, either due to processing delays or the arrival of new information, may be responsible for the revision of the initial decision[9,16]. We note that these results must be considered cautiously due to the relatively low number of CoM trials. This limitation prevented us from reliably identifying post-decisional evidence accumulation-like activity in trials with CoMs as we did for trials with no CoMs in EA-candidate channels. As a result, factors other than evidence accumulation might have contributed to the difference in high gamma activity between trials with and without CoMs. Such factors include sensorimotor signalling, as the onset of CoMs coincided with deviations in mouse trajectories. Importantly, we accounted for motor contributions to our results by re-running population analyses for evidence accumulation and confidence while including instantaneous mouse velocity as a covariate of no interest, with no impact on our main results (see SI−Supplementary Note). We also note that motor aspects should not be treated only as a nuisance, as some are considered key factors of decision-making and metacognitive monitoring[31,39,42,60−62]. Due to this, we are confident that evidence

accumulation drives CoMs in the current task, as was shown in several similar tasks[9,12,16]. Additionally, an alternative account of CoMs proposes that observers may revise their decisions through feedback signalling taking place at a decisional level[63]. Interestingly, we found that activity in the pSMA, which mostly reflected pre-decisional evidence accumulation-like activity, also distinguished trials with and without CoMs, further supporting this hypothesis.

### Single channel level vs regional functional markers

A particularity of our analyses was to combine single-channel analyses with a larger-scale approach, which involved mixed-effects regression models combining all channels within a region of interest. Although these two approaches are mutually informative, we found that the distributions of EA-candidate channels and channels reflecting confidence do not perfectly overlap with the results obtained at the regional scale. These discrepancies may be simply due to the intrinsic limitations of working in a clinical setting with no control over recording sites. However, it could also indicate that the granularity of accumulators' distribution may be finer than the spatial extent of the regions of interest we defined. Indeed, results from single-unit recordings in non-human primates show that neuronal activity is very heterogeneous even within a small area, such as in the lateral intraparietal area[13,64]. Similarly, spatial selectivity in sEEG was found for language or attention[65,66]. This possibility would explain the diversity of results observed in the literature when comparing imaging techniques with lower spatial resolution, such as EEG, and works using high spatial resolution techniques such as single-unit recordings[10,37,67]. It would also explain why we did not find correlates of evidence accumulation at the population level in the parietal cortex, despite considerable literature showing its involvement in evidence accumulation processes[10,68,69]. Finally, we could not find a common population code between evidence accumulation and CoMs using multivariate analyses.

Our results show how evidence accumulation-like activity is found in the cortex on a global scale, in line with recent results indicating that it is an "ubiquitous" process[33,37,41]. Perhaps more importantly, by combining our modelling and electrophysiological results, we uncovered what could be a functional hierarchy of areas supporting confidence. Indeed, we found regions in which evidence accumulation-like activity subserves confidence, either in a pre-decisional manner (e.g. pSMA) or a post-decisional (e.g. insula) manner, as well as regions that did not reflect evidence accumulation but still reflected confidence and CoMs (e.g. OFC). These regions might thus integrate the activity of regions exhibiting evidence accumulation and serve as output systems. Overall, our results provide neurophysiological evidence to explain how a hierarchy of accumulators could provide a diversity of metacognitive processes unfolding at different moments in time.

### Limitations

Our work has several limitations. First, we chose to dichotomise the response process into pre- and post-decisional stages depending on whether it occurred before or after the initial mouse movement. This dichotomy enabled us to interpret our modelling results within a powerful computational framework and rich literature. It also allowed for the operational definition of CoMs. However, it is also reductive since the decision-making process probably extended continuously along the mouse trajectory, leading to the final response. Furthermore, we considered the link between the slope of neural activity and decision times as a functional marker of evidence accumulation. Although this is frequently used[10,70–72], this marker remains debated. Indeed, one could argue that our definition of EA-candidate channels was somewhat circular, as it depended on the correlation between the slope of high-gamma activity and decision times within temporal windows that were themselves defined by decision times. To rule out this potential confound, we have performed additional simulations and analyses

showing that our results were not accounted for by this circularity (see SI−Supplementary Discussion Figs. S12, S13 and S14). Furthermore, our analyses within ROIs, which assessed high-gamma activity as a function of decision time, are not impacted by this confound. Nevertheless, alternatives that consider sensory evidence in addition to decision times could be used in future experiments in which sensory evidence is not titrated to remain around the discrimination threshold. Finally, our macro sEEG recordings did not allow us to assess the differences observed between the pSMA and insula statistically. Implementation schemes varied greatly across patients, and no patient had enough responsive channels in both regions to estimate statistical interactions. Linking cortical activity to fine properties of evidence accumulation, such as the state of winning and losing accumulators, will require single-unit recordings and multivariate analyses in future studies.

## Methods

### Participants

Twenty-four individuals with pharmacologically intractable epilepsy undergoing treatment at the University Hospital in Grenoble, France, were recorded, among which 21 were included in the present article (8 females, mean age = 31.90, sd = 9.78, see Table S4). Patients were treated with a cocktail of two to three antiepileptic drugs selected from the following, listed in order of decreasing frequency of prescription: Carbamazepine, Clobazam, Lacosamide, Levetiracetam, Perampanel, Zonisamide, Lamotrigine, Brivaracetam, Eslicarbazepine, Felbamate, Oxcarbazepine and Topiramate. No statistical method was used to predetermine sample size. Patients were implanted with semi-rigid linear electrodes in preparation for the surgical resection of the seizure focus. The experimental protocol was approved by an ethical committee (MapCog_SEEG 2017-A03248-45), and informed consent was obtained from each patient. The electrode locations were decided following pre-surgical MRI and were based solely on clinical criteria. The recording sites varied across patients, with an average number of 138.8 channels (sd = 27.87). The experiment was performed on a laptop while the patients sat in bed. One participant was excluded due to missing experimental triggers, and two due to excessive epileptic activity. Four additional patients were excluded from the confidence-related analyses due to a failure to correctly use the confidence scale. They were however, included in analyses focusing only on evidence accumulation. Fourteen patients with more than 10 CoM trials were included in the analyses related to CoMs. Thirteen healthy subjects were also included as a control group for behavioural analysis and model-fitting. The data collection for these subjects was approved by the Ethics Committee Sud-Méditerranée II under reference 217 R01, and informed consent was obtained from each control participant.

### Procedure

The task consisted of a random-dot kinematogram 2-AFC discrimination task followed by a confidence judgement. It was implemented under MATLAB R2019b using Psychtoolbox-3[73,74] and ran on a Dell Precision 7550 laptop under Ubuntu 20.04. The task was performed at the hospital with the patients sitting in their beds and the laptop placed on a tablet in front of them. In the visual discrimination task, patients were asked to decide whether a cloud of dots circumscribed to a 108 px radius circle was moving rightward or leftward. Responses were given by moving a computer mouse from its initial position at the bottom centre of the screen to one of two targets presented in the top right and left corners of the screen (108 px radius), corresponding to right and left decisions. Patients were instructed to start moving only after reaching a decision, and were not given any particular instructions with regard to CoMs. Trials were self-initiated by clicking on a $90 \times 19$ px box situated at the bottom of the screen. This step was introduced to ensure that patients would bring back the mouse to a similar starting position at the beginning of every trial. They were instructed to inspect the stimulus for as long as necessary within a limit

of 6 s. If they failed to answer within the 6 s timeframe, a buzzing sound was played and the trial was excluded from the analysis. No feedback was provided to the patients regarding accuracy after each trial. The second part of the task consisted in giving a confidence judgement on the accuracy of the decision, on a visual analogue scale ranging from 0 (sure of having provided an incorrect response) to 100 (sure of having provided a correct response). Patients were encouraged to use the continuous scale as accurately as possible. To minimise the confounding effect of visual-discrimination task difficulty on confidence judgements, motion coherence was titrated to reach 71% of correct responses for every participant using a 1-up/2-down adaptive staircase procedure preliminary to the experiment (80 trials without confidence judgements task). The staircase procedure was maintained throughout the main experiment to account for training or fatigue effects on task performance.

### sEEG data collection and preprocessing

sEEG data were collected during the course of the experiment using semi-rigid linear electrodes (Dixi Microtechniques, Besançon), with a sampling rate of 512 Hz using a Micromed recording system (Micromed, Treviso, Italy). All preprocessing steps were performed in Matlab (R2019b) using the FieldTrip toolbox[75]; version 20211016). Broadband signals from each channel were first visually examined to remove high levels of epileptogenic activity (17.01% of total channels). We then performed bipolar re-referencing among the remaining channels, excluding all channels that did not have a near neighbour among the artefact-free channels. Following bipolar re-referencing, the broadband signal was filtered to extract high-frequency bands between 70 and 150 hz. Seven half-overlapping frequency bands were extracted from the continuous signal in steps of 20 hz. Each frequency band was then normalised, and all bands were averaged. The resulting normalised high-gamma activity was then split into epochs corresponding to 500 ms before stimulus onset up to the end of trials following confidence judgements. Finally, all frequency bands were baseline-corrected using the mean activity in the 500 ms preceding stimulus onset, averaged over time and trials and smoothed using a 400 ms second-order Savitzky-Golay filter. Following filtering, individual trials were visualised, and trials still exhibiting important epilepsy-related artefacts were excluded (7.45% of trials were removed during these steps).

### Statistical analysis

All the analyses described in this section were performed under Python 3.8.8 in a Jupyter Lab environment. All descriptive analyses were performed using Numpy and Scipy[76,77]. All generalised linear models were fitted using StatsModel[78], and mixed generalised linear models were computed with lme4[79] run in Python using Rpy2. Glass-brain visualisations were generated using MNE-Python[80]. For ROIs all analyses were performed on continuous decision times and confidence ratings. We applied a median split to represent average decision times and confidence ratings in two distinct bins only in the figures for graphical purposes.

**Behavioural analyses.** Trials with decision times shorter than 100 or longer than 2500 ms were excluded from the analysis (17.02% of trials), as well as trials with a CoM (see below). To evaluate the relationship between response accuracy and decision time or confidence judgements, we used mixed generalised linear models with a gamma distribution and log link function, as both decision times and confidence were positive with skewed distributions. We used the following model formula:

Decision Time ~ Accuracy + (1|Participant) or Confidence ~ Accuracy + (1|Participant)

CoMs were detected based on crossings of the screen's vertical midline occurring after at least one-third of the vertical distance to the target had been travelled. The onset of a CoM was determined by finding the time at which the mouse cursor velocity reached a minimum prior to this crossing (Fig. 6A).

**Evidence accumulation model.** The model instantiates two anticorrelated accumulators, one for each possible decision outcome[12,16,26,31,49,70], and the confidence judgements were simulated by extending the evidence accumulation process after the initial decision. The free parameters were the rate at which sensory evidence is accumulated (drift rate), the boundary at which accumulated evidence leads to a decision (bound) and the proportion of response time that does not correspond to evidence accumulation (non-decision time). The noise for each accumulator could be assumed to be independent ($\rho = 0$) but previous studies assumed that it has negative cross-correlation which make the two accumulators inhibit each other (e.g. Kiani et al.[12].; Van den Berg et al.[16]). Therefore, to reduce the degrees of freedom of the model, we fixed the covariance between accumulators to $\rho = -\sqrt{0.5}$, to match previous works[16]. Similarly, we fixed the standard deviation of the non-decision time to 60 ms. Although confidence ratings were obtained on a continuous scale during the experiment, they were discretized into two bins to facilitate model fitting. We assumed that patients had high confidence when the difference between the winning and losing accumulators[48,49,81] *tconf* seconds after the decision boundary exceeded a confidence threshold thrconf. To reflect that confidence was defined as a read-out of the decisional process, the model was fitted sequentially, first focusing on decision accuracy and decision times, and then on confidence[9,15,16,18,31,39]. This approach allowed us to test whether a model reproducing decisional features was sufficient to generate suitable confidence readouts, with no trade-off between goodness of fit for decision performance and confidence.

To fit the model, we simulated 1000 trials for each participant with decision times and response accuracies. The sign of the decision time was inverted for simulated errors, allowing us to estimate the log-likelihood based on a Kolmogorov-Smirnov test between the simulated decision time distribution and those observed in the data[82]. We fitted the free parameters with a Nelder-Mead simplex optimisation procedure[83]. To avoid local minima, we initialised the procedure with a wide range of different starting points ($N = 24$). To fit confidence, we used the parameters from the best decision model and simulated confidence ratings based on a readout of post-decisional evidence accumulation for 1000 trials. We computed log-likelihoods as the sum of two components. First, a binomial log-likelihood between the simulated and observed proportion of high-confidence for correct and erroneous responses. Second, a log-likelihood based on a Kolmogorov-Smirnov test between the simulated decision time distribution and those observed in the data for each confidence level and accuracy (correct or error). We optimised the confidence threshold (thr_conf) and readout time (t_conf) free parameters similarly to the decision stage, with 11 readout times ranging from 0 s post-decision (i.e. decisional readout) to 1 s post-decision. For the decisional readout, we set t_conf = 0 and only optimised thr_conf. We took the best model in terms of log-likelihood. To simulate neural data, we assumed that evidence accumulation would (i) stop at the decision (Figs. 5A and S6A), (ii) continue post-decisionally up to the confidence readout (Fig. S6B), or until the end of the trial(Figs. 5B and S6C). We also assumed that the non-decision time would be split between a pre-accumulation non-decision time of 100 ms and the remaining non-decisional time would be a post-decision non-decision time corresponding to motor delays. The extent of this split slightly affected when confidence-related activity would begin prior to the movement onset.

Finally, for CoMs, we set a threshold on the difference between the winning and losing accumulators during the 2 s post-decision to reproduce the proportion of CoMs. Thus, an increase in the activity of the losing accumulator following the initial decision could bring the

simulated neural activity to cross the threshold, leading to a CoM. We then derived the timing of simulated CoM from the threshold crossing times, which we compared to observed CoM timings out-of-sample to avoid overfitting.

**Trial segmentation.** We focused sEEG analyses on two temporal windows of interest aimed at capturing perceptual and decisional processes. The first window was centred on stimulus onset, which was considered to precede evidence accumulation by a few 100 ms. The second window was centred on mouse decision time, which we used as a proxy for decision time. Taking the decision time as a proxy for decision time rather than the time at which patients clicked on the left or right target had the advantage of removing the variability induced by different mouse trajectories between patients. All stimulus-aligned analyses were performed from 200 ms before to 1000 ms after stimulus onset, and all movement-aligned analyses were performed from 500 ms before to 500 ms after decision time. Similarly to the behavioural analyses, trials with decision times inferior to 100 ms and superior to 2500 ms were excluded. Trials leading to an incorrect answer were also excluded, and trials containing CoMs were analysed separately.

**Channel selection.** Only channels in grey matter located in a subset of regions of interest were included in the analysis. These ROIs were defined according to the literature as regions likely to support evidence accumulation and/or confidence judgements[15,20,28–31]. Seven ROIs were selected based on coordinates from the MarsAtlas[84]: visual cortex, parietal cortex, pre/supplementary motor cortex (pSMA), dorso-lateral prefrontal cortex (dlPFC), inferior frontal cortex (IFC), orbito-frontal cortex (OFC), and Insula (see Table 1 in SI - Supplementary Note for details). Of note, some of the deeper channels in intracranial electrodes targeting the IFC were in fact located in the insula. All channels in these electrodes were manually checked against the patients' MRI to ensure that they were correctly labelled.

All analyses were performed after downsampling the original 512 Hz high-gamma signal to 64 Hz to decrease computation time and multiple comparisons. We additionally restricted the analysis to channels considered responsive to the stimulus. To do so, we selected channels for which the average high-gamma activity in a window covering 200 ms post-stimulus to median decision time was significantly superior to the activity during a 500 ms baseline window immediately preceding stimulus onset (trial-averaged high-gamma activity, one-sided $t$ test). A subset of 102 channels was found to be stimulus-sensitive across all patients.

**Evidence accumulation-candidate channels.** Channels exhibiting steeper high-gamma ramps following stimulus onset for shorter decision times were considered as reflecting evidence accumulation[10,35]. We evaluated this relationship by fitting a linear function to single-trial high-gamma activity from stimulus onset to decision time, and correlating the resulting slopes with decision times (Spearman rank order correlation test). Among the 102 responsive channels, 41 were found to have a negative correlation between slopes and decision times. Permutation testing was performed by shuffling the decision times, performing the correlation again with shuffled labels and counting the resulting selected channels, giving us an estimate of the probability of finding our results by chance. This process was repeated 1000 times, and the p-value was computed as $(C + 1) / (n\_perms + 1)$ where C is the number of permutations whose score is superior or equal to the true score.

**Single channels reflecting confidence judgements.** Channels reflecting confidence were selected based on the correlation between high-gamma activity in a 400 ms window preceding or following decision time and confidence judgements. The average activity in these windows of interest for each trial was correlated with confidence judgements (Pearson Rank Correlation test, threshold $p$ value: 0.05).

Channels exhibiting a significant positive correlation between confidence and high-gamma were selected. This analysis was performed on all responsive channels, and the selected channels were labelled either mixed sensitivity channels if they had also been labelled as accumulating evidence, or pure confidence channels if it was not the case. These results were validated using permutation tests on shuffled confidence judgements, with the same procedure as described in the section above.

**Generalised mixed-effects linear models.** We used mixed generalised mixed-effects linear models (mGLMs) to analyse pooled channels across patients for each ROI. Pooling the activity from several channels across patients can represent a challenge, as each participant contributes differently to the overall dataset according to idiosyncratic implantation sites. mGLMs solve this challenge by accounting for the hierarchical nature of this type of data and allowing contributions with distinct weights to the model's predictions. We used nested mGLMs with a gamma distribution and log link function of the following forms:

Hga ~ Decision Time + (1|Participant/channel)

Hga ~ Confidence + (1|Participant/channel).

Random intercepts were fitted for each participant and each channel within each participant. Random slopes could not be added as they induced convergence failures. We fitted one model per data point in the 200 ms–1000 ms window following stimulus onset and a −400 ms to 400 ms window around decision time. $P$ values were obtained for each time point in these windows using a Wald test procedure, and were then corrected for multiple comparisons using false discovery rate[85,86].

Additionally, we ran post hoc analyses to rule out motor confounds considering instantaneous mouse velocity Vel(t) as a covariate (see SI−Supplementary Note Table 1):

Hga ~ Decision Time * Vel(t) + (1|Participant/channel)

Hga ~ Confidence * Vel(t) + (1|Participant/channel)

**Changes of Mind.** Trials with CoMs were analysed separately, and due to their low number, we conducted statistical analyses on all channels and trials across patients for each ROI. Patients with <10 CoMs trials were excluded from the analysis (14 patients were kept). We compared three conditions: CoM trials aligned on the CoM onset, regular trials aligned on decision time, and regular trials aligned on CoM-like onset. This latter condition was to ensure that the effects observed in CoM-trials were not due to another process common to all trials. To obtain CoM-like timing, we first fitted the distribution of CoM times for each participant with a Gaussian density function. We then randomly drew as many timings from this distribution as there were non-change of mind trials and used those timings to align the high-gamma signal. This procedure was repeated 50 times to test the robustness of our findings. To compare the effect of the different conditions over time, we used a generalised linear model of the form Hga ~ Condition with a gamma distribution and log link function. The model was applied to every time point in a −400 ms to 400 ms window, and corrected for multiple comparisons with the false-discovery rate procedure[85,86].

**Reporting summary**
Further information on research design is available in the Nature Portfolio Reporting Summary linked to this article.

## Data availability
The behavioural and high-gamma activity data generated in this study have been deposited in the OpenNeuro database (https://doi.org/10.18112/openneuro.ds006253.v1.0.1). The raw neural data can be obtained upon request by contacting the corresponding author.

## Code availability
Analysis and modelling scripts are publicly available on the Open Science Framework (https://doi.org/10.17605/OSF.IO/2KT97).

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

## Acknowledgements

The authors would like to thank Clarissa Baratin, Blandine Chanteloup, and Manik Bhattacharjee for their help with the electrode localisation procedure, and Emmanuelle Marmet, Marine Carmona for their help with data collection. NF has received funding from the European Research Council (ERC) under the European Union's Horizon 2020 research and innovation programme (Grant Agreement No. 803122). Co-funded by the European Union (ERC, LEAP, 101077874, Volta, 101125379). Views and opinions expressed are however, those of the author(s) only and do not necessarily reflect those of the European Union or the European Research Council. Neither the European Union nor the granting authority can be held responsible for them.

## Author contributions

N.F. and M.P. developed the study concept. N.F. implemented experiments. D.G., F.S., L.G., M.R., and N.F. collected data. D.G., L.G., M.P., and N.F. analysed data. D.H. performed surgeries. L.M., P.K., and A.R. provided clinical data. D.G., N.F., and M.P. drafted the paper. All authors provided critical revisions and approved the final version of the paper for submission.

## Competing interests

The authors declare no competing interests.
