## [Transparent Peer Review file · Nature Communications]

Evidence accumulation in the pre-supplementary motor area and insula drives confidence and changes of mind

Corresponding Author: Dr Dorian Goueytes

Version 0:

Reviewer comments:

Reviewer #1

(Remarks to the Author)

In the manuscript titled "A functional overlap between evidence accumulation, confidence, and changes of mind in the pre-supplementary motor area and insula" by Goueytes et al. the authors analyze a perceptual discrimination and confidence task using intracranial electroencephalography recordings. They find evidence of evidence accumulation in the pre-supplementary motor area and insula, which also accounts for changes of mind and confidence. I find this article to be of high quality, but I have some comments on aspects that I did not fully understand or that I believe could be improved.

1. The use of only two bins for confidence is unusual in the field, as it reduces the resolution richness despite adding computational complexity. Expanding the number of bins could provide more detailed insights.
2. The fitting of the models is done sequentially rather than simultaneously. Wouldn't it be more appropriate to fit accuracy and confidence together to ensure a more integrated analysis?
3. The definition of changes of mind seems imprecise. This appears to be the weakest part of the work, as it only reports greater gamma amplitude in those who have changes of mind. A more precise definition and explanation would strengthen this section.
4. In Figure 3, the color labeling is confusing or lacks sufficient labels for clarity.
5. It might be mentioned, but I am not sure; it would be beneficial to clarify the correction for multiple comparisons multiple times, as many comparisons are made.
6. The authors attempt to distinguish the effects of confidence and decision time by stating: "To ensure that these effects were not due to the interaction between confidence and decision time, we ran a similar mixed-effects regression including an interaction term between confidence and decision time." However, I do not understand how the interaction would reflect that the effect depends on confidence and not reaction time, as these variables covary. It is not the interaction but the contribution of each that discriminates the effects. This should be better explained.
7. "All analyses were performed on the high-gamma signal downsampled from 512 Hz to 64 Hz" is confusing. How could the authors investigate frequencies from 75 to 150 Hz, the high-gamma range, with a sampling rate of 64 Hz?

Overall, I believe this is a strong manuscript with significant findings, but addressing these points would improve its clarity and impact.

(Remarks on code availability)

Reviewer #2

(Remarks to the Author)

Goueytes et al. use a combination of behavior, modeling and sEEG in epilepsy patients to examine the dynamics of decision formation and confidence in several brain regions. The study is timely, well designed and with solid rationale, and the manuscript is well written. Some of the results are quite interesting and potentially important for the field to know about, especially regarding the insula.

However, one of the key claims rests on a pretty subtle effect (a brief blip of pre-movement confidence-related activity in parietal cx and pSMA), and there is reason to be skeptical that even the stronger effects in the insula are attributable to post-decision accumulation, another central claim of the paper. Lastly, the premise that many of the recorded signals reflect evidence accumulation to begin with, as opposed to other processes with superficially similar dynamics, is far from ironclad. See below for more detail on these concerns, followed by a few minor comments.

General comments:

1.
The pre-movement confidence effects in pSMA (Fig. 5A) and parietal cx (S3A) are critical to the claims of the paper but are not very convincing, being rather small in magnitude (relative to the error bars), and brief in duration. Statistical significance notwithstanding, it's hard to know what to make of a difference of ~5% in high gamma power, especially without other lines of evidence supporting the idea that this signal reflects the difference between accumulators (each of which might be expected to reside largely in the contralateral hemisphere, at least for parietal cx).

I was also confused whether the authors' interpretation is that these blips correspond to post-decisional evidence accumulation, as mentioned in Results and Discussion (p. 15), or "pre-decisional confidence", as mentioned when contrasting this with post-decisional confidence in the insula (p. 16).

2.
The main insula result (Fig 5B) is much clearer, but in my view it does not clearly support the claim of "a link between confidence and post-decisional evidence accumulation." Certainly there is an interesting correlate of confidence here, but it does not follow that the observed signal reflects post-decision EA, even granting the classification of these channels as reflecting EA in the first place (but see below). This point is a logical one, but empirically, the post-movement traces (5B) also look mostly flat, unlike the up-then-down pattern in the simulation. In fact, to generate traces that even remotely resemble the data, at least in terms of the duration of separation, the simulated post-decisional accumulation period was extended to 500 ms whereas the confidence readout time inferred from behavior was 300 ms.

3.
More fundamentally, despite impressive progress in this area, I think it's still prudent to scrutinize the assumption that ramping activity inversely correlated with movement onset time is sufficient to demonstrate evidence accumulation. This pattern could be consistent with many alternatives, including an average of step-like state transitions on individual trials (Latimer et al. 2015), a stimulus-independent urgency signal (Cisek et al 2009), or other sensorimotor timing-related signals (eg Leon & Shadlen, Schneider & Ghose 2012).

Examples of additional criteria that could bolster the case include: (i) build-up rate that also varies in graded fashion with the strength of evidence (Kelly & O'Connell, Wyart et. al., Philiastides et al.), (ii) convergence of activity to a common level just before* movement onset (Roitman & Shadlen, Kelly & O'Connell), (iii) increased activity triggered on discrete packets of evidence (Huk et al., 2005; Hanks et al. 2015 & Scott et al. 2017), and (iv) patterns of variance and autocorrelation expected for a bounded accumulation process (Churchland et al., 2011; Steinemann et al. 2023).

[*I could be wrong but it appears that the peak times of the fast- and slow-RT traces in pSMA and parietal (~400-500 ms; Fig. 2C top) are not well matched with the RT distribution of participants (Fig. 1C; mean=1.43s), so criterion (ii) seems unlikely to be met.]

The authors concede that "it is unclear what feature of evidence accumulation was reflected by high gamma activity observed in the pSMA," suggesting it may be the difference between accumulators. A more (or at least equally) parsimonious interpretation may be that it does not reflect evidence accumulation at all.

4. (general but more minor)
The modeling section could be a stronger contribution if it included fits to individual participants, and provided more detail beyond just overall percent correct and mean conf rating conditioned on accuracy. Specifically, it would be useful to see choice/RT/conf as a function of coherence (I realize the staircase procedure means there are not many coherence levels, but there should be at least three), and confidence as a function of coherence for error vs. correct trials. It's of interest to the field how well the hypothesized readout for confidence (balance of evidence after a post-decision period) fits empirical data, and it's also worth checking whether confidence-conditioned accuracy and neural effects persist when controlling for variability in the stimulus (if possible).

Minor/specific comments:

"These temporal profiles are reminiscent of early and late accumulation profiles (Morito & Murata, 2022; Msheik et al., 2022)" -- is somewhat cryptic, I had to look up those papers to know what this meant.

Fig 2C: is it intended to have two panels for pSMA?

The shading (95% CI) is missing in Fig. 6C, left, orange.

Was feedback given (correct/error) after each trial?

What are the units for the y-axis in the high-gamma figures?

The difference between 5A and 5B, lower panels, is not mentioned in the figure legend.

(Remarks on code availability)

Reviewer #3

(Remarks to the Author)

This article uses a unique combination of intracranial human recordings, mouse-tracking, and computational modeling to study evidence accumulation, confidence, and changes of mind in the brain. This is a very ambitious project with an invaluable dataset. I think that some form of this paper merits publication in a top-tier outlet like this one. That being said, the current version of the article suffers from a few important issues. I think that these issues could be addressed with a revision that improves and/or clarifies some analyses and reconsiders some interpretations of the data.

(1) A key analysis in the article is the search for evidence accumulation channels by looking for channels where decision time is negatively correlated with the slope of the increase in activity - they report 41 channels that exhibit this correlation. However, the slope is defined as the rise in activity divided by the decision time. So, there is an inverse correlation between the slope and decision time that is mechanically built into the analysis. Instead of using the slope over the whole decision time, I would instead suggest using the slope over the first XXX ms, where XXX is short enough to be within the decision time for most (e.g. 90%) of the trials, or some other sensible amount of time.

(2) There is a critical assumption throughout the article that the decision occurs when the mouse first starts moving and that everything after that is "post decision". This goes against much of the literature using mouse tracking to study decision making. The assumption is usually that participants start moving their mouse prior to making their decision and that movements along the way reflect the noisy decision making process. So why do the authors think that in their case the mouse movements are post-decision? Doesn't the fact that there are "changes of mind" contradict the notion that participants have made their decision when they start moving their mouse? To me, this "post-decision" interpretation felt post-hoc.

An alternative interpretation is that the participant makes a first decision to start moving their mouse once they are reasonably confident in an answer (which quite sensibly occurs in preSMA), but then they continue to evaluate the stimulus and gather evidence the rest of the way (which seems to occur in the insula). This is similar to the authors' existing story but does not require the strong assumption that people are continuing to decide after the decision has been made. As a side note, I think it is worth noting that the pre-SMA and insula both exhibit a substantial peak in their regression coefficients 0.4s "post decision", suggesting that the pre-SMA is indeed still involved during that period. I would be wary of drawing strong conclusions about the difference between those pre-SMA and insula time courses based on Figure 4, especially given the sample size.

(3) I wonder about the authors' confidence model. Their model is that confidence is based on the amount of accumulated evidence 0 to 1s after the decision. Again, I question whether this activity is really occurring after the decision. If that were the case, it would imply that people would be unable to report their confidence when queried immediately after their decision. This does not seem plausible. Instead, if you think of that time as simply a slice of their decision time, then what you are measuring is the drift rate, which should indeed correlate with confidence. It would be useful to know whether this 1 s time window correlates more strongly with confidence than say the 0 to 1s window after stimulus onset. One would expect that to be true if the first few hundred milliseconds are noisy due to initial stimulus encoding, i.e. non-decision time. But what about the interval from say 0.5s to 1.5s post stimulus onset? How does that compare?

(4) Could the authors provide more concrete detail about the decision model? What does it mean that the accumulators are anti-correlated, and why this particular covariance factor of square root of 0.5?

(5) Regarding exclusions - the authors exclude a lot of trials (17%). Why give participants 6 s to decide but then exclude trials with decisions longer than 2.5 s? Are the results robust to these exclusions? I also didn't understand the comment about how these longer decisions had "noisy trajectories", since the trajectories occur after the decision time. Why would they be related? If anything, I would think that trajectories would need to be more ballistic if a participant spent too long leaving the start point.

Other more minor comments:

I didn't understand the point on p.7 where the authors state "Because of this low number of identified channels related to error processing, we focused the rest of the analysis on correct trials." Why should error trials be excluded? From the participant's perspective they don't know the difference between correct and error. Error trials are simply on the more difficult

end of the spectrum than the correct trials. Why do the authors need to throw away a quarter of their data?

Because the authors were using a staircase procedure to adjust the difficulty, do they have a trial-by-trial measure of the dot motion coherence? Could they use this variable to test whether the purported evidence accumulation regions had steeper slopes on the higher coherence trials, that participants were more confident on higher coherence trials, and that there were fewer changes of mind on higher coherence trials?

In a couple of places the authors try to argue that what they're seeing is not just a correlation with decision time. On page 9 they state that "we verified that the magnitude of the correlation between high-gamma activity and decision time was significantly lower than that of the correlations between high-gamma activity and confidence." I'm not sure what this is supposed to tell us. This difference in correlation could be due to the fact that decision time is more noisy than the confidence measure, or because the relationship with confidence is more linear than with decision time. Later, they talk about interacting confidence and decision time. I did not understand the motivation behind this analysis or what it tells us about the relative contributions of confidence and decision time.

Given some of the statements about OFC being involved in the output of the decision, I was surprised not to see any reference to the work on value based decision making, such as Camillo Padoa-Schioppa's recent work.

Ideally the mixed-effects regressions should include random slopes as well as intercepts to account for all sources of across-participant variability.

At one point in the paper the authors state that "These assumptions did not have much influence on our conclusions." This statement is too vague. To what extent did the authors explore different analyses/parameters?

Figure 1A - there is a reference to dark gray channels that were not selected for analysis but I do not see them in the figure.

Figure 3 - why isn't the green/gray distinction in the figure legend rather than in the caption? This was a strange departure from the other figures.

(Remarks on code availability)

Version 1:

Reviewer comments:

Reviewer #1

(Remarks to the Author)

all my concerns have been addressed, congratulations to the authors

(Remarks on code availability)

Reviewer #2

(Remarks to the Author)

I thank the authors for the clear and rigorous response to reviewer comments, and congratulate them on a very nice study. Although I still find the effect sizes to be modest (and brief in pSMA), I do appreciate the method-specific conventions and statistical issues, as well as the challenges involved in collecting this valuable dataset in a hospital setting. The added details and caveats about how to interpret EA-like signals in EEG data adequately address my concerns in the sense that it can be left up to the field to decide, and this paper provides a worthy addition to a growing and exciting literature.

I have only one minor comment: wasn't sure about the header, "Response accuracy, decision times, and confidence *as* accumulated evidence" -- do you mean 'vs' or 'explained by' or 'as a function of'?

(Remarks on code availability)

Reviewer #3

(Remarks to the Author)

The authors have addressed almost all of my concerns. However, they failed to adequately address my #1 concern, which was about the correlation between slope of the increase in activity and decision time. I pointed out that slope is computed as the rise in activity divided by time, and so you are correlating (activity/time) ~ time. There is a mechanical correlation here

because decision time is on both sides of the equation.

Consider the following possibility - activity rises to the same level with the same slope in every trial and stays there until the end of the trial, independent of the decision time. In trials with longer decision times, this would equate to a smaller slope than in trials with shorter decision times. As a result, you would find a negative correlation between slope and decision time, as the authors observe in those 41 channels.

I don't see how a permutation test addresses this issue. A permutation test simply tells us that when you shuffle the decision times on the right hand side of the correlation, the correlation goes away, as it would even if the correlation were mechanical.

This issue needs to be addressed more thoroughly/convincingly.

(Remarks on code availability)

Version 2:

Reviewer comments:

Reviewer #3

(Remarks to the Author)

The authors have addressed my remaining concerns. I'm now happy to endorse this article for publication.

(Remarks on code availability)

Color code: The Reviewers' comments are in black, our responses in red, and existing or updated segments from the manuscript are highlighted in yellow

Reviewer #1 (Remarks to the Author):

In the manuscript titled "A functional overlap between evidence accumulation, confidence, and changes of mind in the pre-supplementary motor area and insula" by Goueytes et al. the authors analyze a perceptual discrimination and confidence task using intracranial electroencephalography recordings. They find evidence of evidence accumulation in the pre-supplementary motor area and insula, which also accounts for changes of mind and confidence. I find this article to be of high quality, but I have some comments on aspects that I did not fully understand or that I believe could be improved.

We thank the reviewer for recognizing the quality of our work and for raising important points, which we address below.

1. The use of only two bins for confidence is unusual in the field, as it reduces the resolution richness despite adding computational complexity. Expanding the number of bins could provide more detailed insights.

We agree that using two bins for confidence is suboptimal. In fact, we used continuous decision times and confidence in all behavioral and neural data analyses. We only binarized decision times and confidence for illustration purposes when plotting results. This was made explicit for decision times in the legend of Figure 2 ("Even if statistics were performed on continuous decision times, and for illustration purposes only, blue lines correspond to averaged high-gamma for the 50% fastest trials and light grey lines the 50% slowest."). We now made this clear also for confidence in Figures 4-5, as well as in the methods section: "All analyses were performed on continuous decision times, and binarized only for illustrative purposes."

The figures below represent high-gamma activity in the pMSA and Insula depending on confidence tertiles, similar to our initial findings.

2. The fitting of the models is done sequentially rather than simultaneously. Wouldn't it be more appropriate to fit accuracy and confidence together to ensure a more integrated analysis?

In our view, the fact that accuracy and confidence were fitted sequentially does not mean they are not integrated or that confidence neglects the first-order evidence underlying task accuracy. As we describe in the methods section and previous work (Faivre et al., 2020; Pereira et al., 2020), the sequential nature of this approach is inherent to the fact that we consider confidence as a readout of the decisional process. As our goal was to compare confidence rather than decision-making models, we decided to fix the decisional stage. Indeed, for comparison purposes, we wanted to avoid models from trading worse fits of the decision stage (RT and choice) for better confidence fits. Although some works have used a simultaneous fitting procedure (Pleskac & Busemeyer, 2010; Rausch et al., 2020), the sequential procedure we used is widespread in the field (Desender et al., 2021; Faivre et al., 2020; Pereira et al., 2020, 2021; Resulaj et al., 2009; van den Berg et al., 2016). We updated the methods section to justify our fitting procedure choice better.

To reflect that confidence was defined as a read-out of the decisional process, the model was fitted sequentially, first focusing on decision accuracy and decision times, and then on confidence (Desender et al., 2021; Faivre et al., 2020; Pereira et al., 2020, 2021; Resulaj et al., 2009; van den Berg et al., 2016). This approach allowed us to test whether a model reproducing decisional features was sufficient to generate suitable confidence read-outs, with no trade-off between goodness of fit for decision performance and confidence.

3. The definition of changes of mind seems imprecise. This appears to be the weakest part of the work, as it only reports greater gamma amplitude in those who have changes of mind. A more precise definition and explanation would strengthen this section.

As stated in the methods section, CoMs were detected based on crossings of the screen's vertical midline occurring after at least one-third of the vertical distance to the target had been travelled. The onset of a CoM was determined by finding the time at which the mouse cursor velocity reached a minimum prior to this crossing. We updated Fig.6A to make this more explicit.

Figure 6. Neural Correlates of CoMs. **(A)** Mouse trajectories and velocity profile for regular trials (left) and CoM trials (right) for one example participant. The red circle indicates the onset of a CoM, defined by a change in the mouse direction and velocity. **(B)** Comparison of observed and simulated CoM onset. Blue line shows simulated data. Grey histogram corresponds to observed CoM-onsets binned and averaged across subjects. **(C)** CoM-aligned high-gamma activity for CoM trials compared to high-gamma activity (in arbitrary units) aligned on CoM-like times for non-CoM trials (left panel) or move-aligned high-gamma for non-CoM trials (right panel). Dark bars indicate a significant effect of condition (CoM/non-CoM) on high-gamma levels ($p < 0.05$, FDR corrected using the Benjamini-Hochberg method). Shaded areas correspond to 95% CI.

Besides this clarification of definition, we would like to stress why our findings about CoMs are valuable. First, they explain CoM onsets mechanically using a computational evidence

accumulation model. Second, they link the occurrence of CoMs with high gamma activity in two brain regions already found to reflect evidence accumulation. Third, they are highly novel since they derive from the first joint recording of sEEG and mouse tracking. Nevertheless, we acknowledged in the original manuscript that CoM data are inherently noisy, representing only ~10% of total trials:

Results: "We note, however, that the same analysis restricted to channels reflecting evidence accumulation did not yield any significant result, possibly due to lack of statistical power as CoMs were rare".

Discussion: "We note that these results must be considered cautiously due to the relatively low number of CoM trials. This limitation prevented us from reliably identifying the hallmarks of post-decisional evidence accumulation in trials with CoMs as we did for initial evidence accumulation in trials with no CoMs. As a result, factors other than evidence accumulation might have contributed to the difference in high gamma activity between trials with and without CoMs. Such factors include sensorimotor signalling, as the onset of CoMs coincided with deviations in mouse trajectories."

Although we noted the scarcity of data, we should have explained why CoMs were so rare in our study. Other studies imposing a time pressure to initiate a decision typically produce more CoMs. However, these changes of mind under time pressure are less ecological, as they reflect, at least partially, participants making an initial, quasi-random decision to comply with instructions, and then making their actual decision. Our paradigm captured better what occurs when one revises a decision, in the sense that participants who were not under time pressure made a first genuine decision, then changed their mind to make a second decision in the rare trials where they felt their initial decision was erroneous. We added a sentence in the discussion to explain why changes of mind were rare in the absence of time pressure, which may contribute to noisier results:

It should be noted that the CoMs in our study were observed despite the absence of time pressure to respond. In this sense, CoMs do not reflect a situation where participants hastily moved the mouse to one side or the other to comply with the instruction, and then made an actual decision in a second step. On the contrary, the CoMs we observed were likely endogenous, so that the first observed mouse movement reflected a genuine decision rather than a random movement. The lack of time pressure increases the ecological validity, but results in fewer CoMs.

In sum, we have better defined and described CoMs, acknowledged more transparently the noisy nature of the data, explained the scarcity of data due to the absence of temporal pressure, and emphasized the mechanistic character of our conclusions, namely that CoMs

occur when postdecisional evidence accumulation instantiated by the insula reverses to a given threshold. In light of these elements, we hope the reviewer will better appreciate the value of our findings on CoMs.

4. In Figure 3, the color labeling is confusing or lacks sufficient labels for clarity.

This has been corrected.

5. It might be mentioned, but I am not sure; it would be beneficial to clarify the correction for multiple comparisons multiple times, as many comparisons are made.

We systematically relied on permutation tests when selecting channels and used false-discovery rate corrections when interpreting time-wise analyses. We made this more explicit in the methods section and throughout the text:

We performed generalised linear mixed-effects regressions between high-gamma activity and decision times at each timepoint following the stimulus onset (while correcting for multiple comparisons, see Methods) to test this hypothesis.

P-values were obtained for each time point in these windows using a Wald test procedure, and were then corrected for multiple comparisons using false discovery rate (Benjamini & Hochberg, 1995, 2000).

The model was applied to every time point in a -400 ms to 400 ms window, and corrected for multiple comparisons with the false-discovery rate procedure (Benjamini & Hochberg, 1995, 2000).

6. The authors attempt to distinguish the effects of confidence and decision time by stating: "To ensure that these effects were not due to the interaction between confidence and decision time, we ran a similar mixed-effects regression including an interaction term between confidence and decision time." However, I do not understand how the interaction would reflect that the effect depends on confidence and not reaction time, as these variables covary. It is not the interaction but the contribution of each that discriminates the effects. This should be better explained.

We agree with the reviewer that this analysis was not entirely satisfying. We replaced it with a new analysis relying on a full linear mixed-effects regression, including confidence and

decision time as independent variables: (Hga ~ Confidence*Decision Time + (1|Participant/channel)). This new analysis performed on the clusters of interest in the pSMA and the Insula shows a significant effect of confidence on high-gamma activity in the two ROIs, alongside significant effects of decision time for both ROIs in the move-aligned window. This new analysis confirms that high-gamma activity in the pSMA and Insula reflects confidence, even after the variance from decision times is accounted for. We have updated the SI accordingly.

We also tested this potential confound at the population level by running a full linear mixed-effects regression, including confidence and decision time as independent variables as well as their interaction, on the clusters of interest in the pSMA and the insula with the following formula:

(Hga ~ Confidence*Decision Time + (1|Participant/channel))

7. "All analyses were performed on the high-gamma signal downsampled from 512 Hz to 64 Hz" is confusing. How could the authors investigate frequencies from 75 to 150 Hz, the high-gamma range, with a sampling rate of 64 Hz?

Sorry if this sentence was unclear. As commonly done in the field (Cecchi et al., 2022; Merrick et al., 2022), we first extracted HGA from data sampled at 512Hz, and only then downsampled the high-gamma signal to 64Hz to perform the statistical analysis. We clarified this in the revised manuscript: All analyses were performed after downsampling the original 512 Hz high-gamma signal to 64 Hz to decrease computation time and multiple comparisons.

Overall, I believe this is a strong manuscript with significant findings, but addressing these points would improve its clarity and impact.

We thank the reviewer for their appreciation of our work and for helping us improve our manuscript.

Reviewer #2 (Remarks to the Author):

Goueytes et al. use a combination of behavior, modeling and sEEG in epilepsy patients to examine the dynamics of decision formation and confidence in several brain regions. The study is timely, well designed and with solid rationale, and the manuscript is well written. Some of the results are quite interesting and potentially important for the field to know about, especially regarding the insula.

However, one of the key claims rests on a pretty subtle effect (a brief blip of pre-movement confidence-related activity in parietal cx and pSMA), and there is reason to be skeptical that even the stronger effects in the insula are attributable to post-decision accumulation, another central claim of the paper. Lastly, the premise that many of the recorded signals reflect evidence accumulation to begin with, as opposed to other processes with superficially similar dynamics, is far from ironclad. See below for more detail on these concerns, followed by a few minor comments.

We thank the reviewer for noting the quality of our work and raising these concerns, which we will address fully below.

General comments:

1. The pre-movement confidence effects in pSMA (Fig. 5A) and parietal cx (S3A) are critical to the claims of the paper but are not very convincing, being rather small in magnitude (relative to the error bars), and brief in duration. Statistical significance notwithstanding, it's hard to know what to make of a difference of ~5% in high gamma power, especially without other lines of evidence supporting the idea that this signal reflects the difference between accumulators (each of which might be expected to reside largely in the contralateral hemisphere, at least for parietal cx).

I was also confused whether the authors' interpretation is that these blips correspond to post-decisional evidence accumulation, as mentioned in Results and Discussion (p. 15), or "pre-decisional confidence", as mentioned when contrasting this with post-decisional confidence in the insula (p. 16).

We thank the reviewer for this critical evaluation. We separate our answer into two parts: i) whether our pre-movement correlates of confidence are convincing and ii) whether pre-movement effects should be interpreted as pre- or post-decisional.

i) Whether our pre-movement correlates of confidence are convincing

We agree that further characterising effect sizes found in the pSMA and parietal cortex regarding pre-movement confidence is important. Estimating effect sizes in generalised mixed-effects regression with complex random structures is far from trivial. We attempted to do so by re-fitting a linear mixed-effects model, taking z-scored high-gamma activity as the dependent variable and confidence as fixed effect. We found a standardized beta coefficient of 0.188, similar to what we saw in the Insula (0.193). We now provide this evaluation of effect sizes in the revised supplementary information:

In order to evaluate the comparative magnitude of the significant effect of confidence on high-gamma modulation observed in the pSMA and the Insula, we refitted a linear mixed-effects model, taking standardized high-gamma activity and confidence to obtain standardized coefficients. We obtained similar standardized beta coefficients for the pSMA and the Insula (0.188 and 0.193, respectively), indicating that the effect of confidence observed in these two ROIs is of similar magnitude.

The duration of the observed confidence effect in the pSMA was 187 ms, which is not short—it is almost four times the 50 ms criterion we used to consider an effect. Therefore, we do not think this effect is a blip but a sustained correlate of confidence, with an effect magnitude similar to what we observed in the insula. We would like to add that this effect is not shorter than what is typically found in sEEG, not to mention scalp EEG (Dou et al., 2023; Feuerriegel et al., 2022; Grogan et al., 2023). These noisy empirical observations are also supported by theoretical predictions from a computational model trained on behavioral data only. Thus, the similarity between simulations and observed HGA responses in the pSMA is a solid argument in favor of the role of evidence accumulation in confidence. While many studies have found similar correlates of evidence accumulation with indirect measures of neuronal activity, such as scalp EEG (Dou et al., 2023; Feuerriegel et al., 2022; Grogan et al., 2023; O'Connell et al., 2012) or intracranial EEG (Gherman et al., 2023), it is still unknown how this activity relates to individual accumulators.

Our study does not differ and we made a modeling choice of taking the difference between the two accumulators because many previous models have used this *balance-of-evidence* (Vickers, 1979) rule to compute confidence (Kiani et al., 2014; Moreno-Bote, 2010; van den Berg et al., 2016). Of course, one would need to measure many individual neurons to make a strong connection between spiking activity and the difference between two accumulators, which is still hard to achieve in awake humans performing psychophysical tasks. That is why we did not make strong claims that it is the difference between accumulators rather than the

sum of accumulators: As the state of each of the two accumulators could not be distinguished at the sEEG level (i.e., we could not find univariate or multivariate signatures of rightward vs leftward visual motion), it is unclear what feature of evidence accumulation was reflected by high gamma activity observed in the pSMA. We now make this modeling choice clearer in the discussion:

We considered the balance-of-evidence hypothesis (Vickers, 1979) stating that confidence is proportional to the difference between accumulators (Kiani et al., 2014; Moreno-Bote, 2010; van den Berg et al., 2016). Simulating how this difference between accumulators evolves over time, we could qualitatively reproduce the observed high-gamma activity (see Fig. S11 for alternatives).

We then reasoned that discussing the various possible computational origins of these signals in the discussion was valuable. We removed this section of the discussion and now simply consider the predictions of different models which we show in Fig. S11: the difference of accumulators initially proposed based on the balance-of-evidence hypothesis but also a model using the sum of accumulators (assuming that neural activity from both accumulators sum up) and a model using the winning accumulator (assuming that the choice-incongruent accumulator is disregarded (Zylberberg et al., 2012)). We also further acknowledge that we cannot draw strong conclusions about how different accumulators relate to the observed signal: Linking cortical activity to precise features of the evidence accumulation process will require more research, including cortical recordings of many individual neurons.

Linking cortical activity to fine properties of evidence accumulation, such as the state of winning and losing accumulators will require single-unit recordings and multivariate analyses in future studies.

These modifications should clarify the fact that we do not aim at formally comparing possible underlying computational origins but still consider that using a model that has received extensive behavioral validation is valuable to interpret our neural data. Nonetheless, we agree that for the pSMA, we should be careful in not over-interpreting model simulations. Regarding laterality, the pSMA contained an equal number of channels reflecting evidence accumulation in the left and right hemispheres. We also corrected a typo and now clearly interpret the findings about pSMA as pre-decisional throughout the manuscript.

In sum, using state-of-the-art statistical methods and computational modeling, we found neural correlates of confidence in channels and regions related to evidence accumulation. We believe that the combination of these results supports the paper's claim that "confidence

and CoMs result from EA instantiated before the decision in the pre-supplementary motor area, and after the decision in the insula.”

ii) Whether pre-movement effects should be interpreted as pre- or post-decisional.

We now provide Fig. S6 showing the modelling results for different assumptions on pre- vs. post-decisional evidence accumulation and increasing the number of simulations. While we still consider that activity is incompatible with the type of post-decisional accumulation that could support changes of mind (such as activity in the insula), we agree that we should be careful when interpreting the pre-movement effects as solely pre-decisional. On top of the changes we described above, we also amended the discussion to suggest that a pre-decisional evidence accumulation model could explain the observed high-gamma activity. We thank the reviewer for helping us refine our interpretations.

Interestingly, our model could reproduce the pre-decisional confidence-related activity under the assumption that the signal observed was related to pre-decisional evidence accumulation, but that trials were sorted according to confidence computed post-decisionally (and thus elsewhere in the brain).

Figure S6 - Top. Comparison of model simulations (difference between accumulators) for pSMA activity for **A.** accumulation-to-bound (accumulation stops after a decision boundary is crossed), **B.** accumulation-to-confidence readout (accumulation continues for a few hundred milliseconds until confidence is read out) and **C.** continuous post-decisional accumulation (until the end of the trial). Note that for panel A, one needs to assume that confidence is computed elsewhere in the brain, by post-decisional evidence accumulation lasting until the confidence readout. Left panels: time window locked on stimulus onset, right panels: time window locked on movement onset.

2. The main insula result (Fig 5B) is much clearer, but in my view it does not clearly support the claim of "a link between confidence and post-decisional evidence accumulation." Certainly there is an interesting correlate of confidence here, but it does not follow that the observed signal reflects post-decision EA, even granting the classification of these channels as reflecting EA in the first place (but see below). This point is a logical one, but empirically,

the post-movement traces (5B) also look mostly flat, unlike the up-then-down pattern in the simulation. In fact, to generate traces that even remotely resemble the data, at least in terms of the duration of separation, the simulated post-decisional accumulation period was extended to 500 ms whereas the confidence readout time inferred from behavior was 300 ms.

The reviewer rightly notes that the fact that a channel is identified as reflecting evidence accumulation at an instant t does not necessarily presume that a correlate of confidence observed later within the same channel itself reflects evidence accumulation. We agree with this point, and should not have presented this observation as such. In the revised discussion, we now consider the possibility that “a first insular population of neurons implements evidence accumulation following stimulus onset, and then a second insular population, disjoint from the first, implements the estimation of confidence after the decision”. We consider this possibility as parsimonious as the one mentioned above, but we do not have the means to formally arbitrate between the two with the data we have on hand. In the revised discussion, we note that “Future single-neuron recordings will be needed to confirm that the same population of insular neurons is involved in evidence accumulation and post-decisional confidence”.

Concerning the readout latencies, we chose 500 ms to account for the fact that there should be regions that continue accumulating evidence for changes of mind even after the confidence readout. We now provide simulations using a 1500 ms window that can account for most changes of mind and leads to flatter traces.

Figure. S6 - Bottom Comparison of model simulations (difference between accumulators) for the Insula activity for **A.** accumulation-to-bound (accumulation stops after a decision boundary is crossed), **B.** accumulation-to-confidence readout (accumulation continues for a few hundred milliseconds until confidence is read out) and **C.** continuous post-decisional accumulation (until the end of the trial). Note that for panel A, one needs to assume that confidence is computed elsewhere in the brain, by post-decisional evidence accumulation lasting until the confidence readout. Left panels: time window locked on stimulus onset, right panels: time window locked on movement onset.

3. More fundamentally, despite impressive progress in this area, I think it's still prudent to scrutinize the assumption that ramping activity inversely correlated with movement onset time is sufficient to demonstrate evidence accumulation. This pattern could be consistent with many alternatives, including an average of step-like state transitions on individual trials (Latimer et al. 2015), a stimulus-independent urgency signal (Cisek et al 2009), or other sensorimotor timing-related signals (eg Leon & Shadlen, Schneider & Ghose 2012).

Examples of additional criteria that could bolster the case include: (i) **build-up rate that also varies in graded fashion with the strength of evidence** (Kelly & O'Connell, Wyart et. al., Philiastides et al.), (ii) **convergence of activity to a common level just before* movement onset** (Roitman & Shadlen, Kelly & O'Connell), (iii) increased activity triggered on discrete packets of evidence (Huk et al., 2005; Hanks et al. 2015 & Scott et al. 2017), and (iv) patterns of variance and autocorrelation expected for a bounded accumulation process (Churchland et al., 2011; Steinemann et al. 2023).

[*I could be wrong but it appears that the peak times of the fast- and slow-RT traces in pSMA and parietal (~400-500 ms; Fig. 2C top) are not well matched with the RT distribution of participants (Fig. 1C; mean=1.43s), so criterion (ii) seems unlikely to be met.]

The authors concede that "it is unclear what feature of evidence accumulation was reflected by high gamma activity observed in the pSMA," suggesting it may be the difference between accumulators. A more (or at least equally) parsimonious interpretation may be that it does not reflect evidence accumulation at all.

We thank the reviewer for these suggestions. We agree that ramping activity should be interpreted carefully as a neural marker of evidence accumulation and that alternatives should be considered. We attempted to evaluate the first four alternative explanations suggested by the reviewer, in which sensory evidence serves as a covariate of interest. To do so, we ran new analyses assessing the link between high gamma activity and decision times with sensory evidence as a co-variate: $HGA \sim \text{decision time} * \text{sensory evidence}$. These new analyses performed at the level of regions of interest revealed an effect of sensory evidence only for a short duration in the insula and no effect of interaction between decision times and sensory evidence (see below and revised SI). The main effect of decision times was unchanged in all cases. These analyses are not optimal as sensory evidence was titrated throughout the experiment using a 1up-2down staircase procedure, and therefore did not vary within participants to impact high gamma activity. Moreover, the initial staircasing blocks where sensory evidence varied greatly were not recorded for most participants, preventing us from exploring this issue further. As a result, we cannot formally assess the

alternative models i-iv) proposed by the reviewer. In a new limitation section, we now acknowledge several weaknesses identified by the reviewer, including our inability to formally reject some alternative hypotheses due to the limitations imposed by our experimental design:

[...] Furthermore, we considered the link between the slope of high-gamma activity and decision times as a functional marker of evidence accumulation. Although this is frequently used (Kiani et al., 2008; Mazurek et al., 2003; Ploran et al., 2007; Tremel & Wheeler, 2015), this marker remains debated, and alternatives that consider sensory evidence in addition to decision times could be used in future experiments in which sensory evidence is not titrated to remain around the discrimination threshold.

Figure. S7 Stimulus-aligned and movement-aligned high-gamma activity for regions of interest reflecting evidence accumulation (A, B, C) Top: Stimulus-aligned and movement-aligned high-gamma activity averaged

across participants for all EA channels. Dark bars indicate a significant relationship between high-gamma activity and decision time ($p < 0.05$, FDR-corrected). Red bars indicate a significant relationship between high-gamma activity and motion coherence ($p < 0.05$, FDR-corrected). Blue lines correspond to averaged high-gamma for the 50% fastest trials and grey lines for the 50% slowest. Shaded areas correspond to 95% CI. Bottom: Regression coefficient as a function of time in the stimulus -and movement-aligned window. Coefficients corresponding to the effect of decision time are shown in blue, the effect of sensory evidence in red, and their interaction in purple. Light blue markers indicate the onset and offset times of significant effect of decision time, and the red marker indicates peak effects of decision time.

4. (general but more minor)

The modeling section could be a stronger contribution if it included fits to individual participants, and provided more detail beyond just overall percent correct and mean conf rating conditioned on accuracy. Specifically, it would be useful to see choice/RT/conf as a function of coherence (I realize the staircase procedure means there are not many coherence levels, but there should be at least three), and confidence as a function of coherence for error vs. correct trials. It's of interest to the field how well the hypothesized readout for confidence (balance of evidence after a post-decision period) fits empirical data, and it's also worth checking whether confidence-conditioned accuracy and neural effects persist when controlling for variability in the stimulus (if possible).

We now provide individual fits for choice, decision time, and confidence as supplementary figures (see below). However, the staircase procedure that we used leads to only one level of perceptual difficulty. As noted in our previous answer sensory evidence did not offer sufficient variance within participants for meaningful interpretation and as much as we would have liked to have different levels of coherence, this was not possible to implement in the short timeframes imposed by clinical constraints. We could nonetheless verify that the simulated results are comparable with data obtained in previous studies (Le Denmat et al., 2024; Sanders et al., 2016), confirming the validity of the readout we propose based on previous studies.

Simulation when varying drift rate compared to the fitted drift rate (100% coherence). Left: increasing accuracy obtained when simulating increasing coherence values. Right: Confidence increases with coherence for correct decisions and decreases with coherence for incorrect decisions (Sanders et al. 2016; Le Denmat et al., 2024).

We agree that it is interesting for the field to assess how well the balance of evidence model fits empirical data. The purpose of our model was primarily to provide a mechanistic framework to guide the interpretation of our neural data. It is reasonable to believe that most post-decisional models of confidence would result in similar neural predictions. We now refer the reader to works considering flavours of post-decisional readouts :

We opted for a readout occurring at a fixed latency after the decision (with some variability in the non-decision time), similar to previous works (Pereira et al., 2020; Pleskac & Busemeyer, 2010; van den Berg et al., 2016). We note that other studies have hypothesized that the readout occurs when accumulated evidence reaches additional collapsing bounds (Herregods et al., 2023; Moran et al., 2015), without significantly changing the signatures (Pleskac & Busemeyer, 2010; Sanders et al., 2016) of the so-computed confidence.

Figure. S10.1 Individual proportion of response times for error (red) and correct (green) trials. Barplots show behavioral data and dark traces show model fits.

Figure. S10.2 Individual proportion correct trials (correct direction of movement onset). Barplots show behavioral data and squares show model fits.

Figure. S10.3 Individual proportion of high confidence responses for error (Err.) and correct (Cor.) trials. Barplots show behavioral data and squares show model fits.

Minor/specific comments:

“These temporal profiles are reminiscent of early and late accumulation profiles (Morito & Murata, 2022; Msheik et al., 2022)” -- is somewhat cryptic, I had to look up those papers to know what this meant.

This has been clarified: **These results suggest that evidence accumulation might take various forms, with different functional roles and temporal profiles, as shown previously using fMRI (Morito & Murata, 2022; Msheik et al., 2022).**

Fig 2C: is it intended to have two panels for pSMA?

These two panels represent high-gamma activity for two distinct channels across two different participants. The figure and the legend have been updated to clarify this point.

The shading (95% CI) is missing in Fig. 6C, left, orange.

The shading is present but very narrow. The figure has been updated to make it more visible.

Was feedback given (correct/error) after each trial?

No feedback was given during the experiment, this point has been clarified in the methods section:

No feedback was provided to the participants regarding accuracy after each trial.

What are the units for the y-axis in the high-gamma figures?

High-gamma activity is shown in arbitrary units as it is estimated from an average of seven normalised overlapping frequency bands between 70 and 150 Hz. This has been clarified in the methods section and figure legends.

The difference between 5A and 5B, lower panels, is not mentioned in the figure legend.

We have updated the legend to include this information.

Reviewer #3 (Remarks to the Author):

This article uses a unique combination of intracranial human recordings, mouse-tracking, and computational modeling to study evidence accumulation, confidence, and changes of mind in the brain. This is a very ambitious project with an invaluable dataset. I think that some form of this paper merits publication in a top-tier outlet like this one. That being said, the current version of the article suffers from a few important issues. I think that these issues could be addressed with a revision that improves and/or clarifies some analyses and reconsiders some interpretations of the data.

We thank the reviewer for noting the value of our data and results. We substantially revised the manuscript, clarifying the text in several instances and interpreting some of our results more parsimoniously.

(1) A key analysis in the article is the search for evidence accumulation channels by looking for channels where decision time is negatively correlated with the slope of the increase in activity - they report 41 channels that exhibit this correlation. However, the slope is defined as the rise in activity divided by the decision time. So, there is an inverse correlation between the slope and decision time that is mechanically built into the analysis. Instead of using the slope over the whole decision time, **I would instead suggest using the slope over the first XXX ms, where XXX is short enough to be within the decision time for most (e.g. 90%) of the trials, or some other sensible amount of time.**

We accounted for this important point at the statistical level using permutation tests. We now make it more explicit in the revised results section:

We assessed the significance of this result by repeating the analysis while randomly permutating the decision times with regard to high-gamma activity. This strategy ruled out that the observed slopes stemmed from the inverse correlation between the slope and decision time built into the analysis (i.e., steeper slopes may be found within shorter epochs among trials with short decision times). We obtained significant results (1000 permutations, $p = 0.001$, see methods), indicating that our effect was indeed linked to a ramping-up of high-gamma activity and not a statistical artifact.

Note that due to the spread of the distribution of decision times, it was not possible to find a meaningful time interval to perform the reviewer's analysis without excluding a vast amount of data.

(2) There is a critical assumption throughout the article that the decision occurs when the mouse first starts moving and that everything after that is “post decision”. This goes against much of the literature using mouse tracking to study decision making. The assumption is usually that participants start moving their mouse prior to making their decision and that movements along the way reflect the noisy decision making process. So why do the authors think that in their case the mouse movements are post-decision? Doesn't the fact that there are “changes of mind” contradict the notion that participants have made their decision when they start moving their mouse? To me, this “post-decision” interpretation felt post-hoc.

We thank the reviewer for this suggestion, which might stem from a need for more clarity on our part regarding the instructions given to participants. As explained in response to a comment made by reviewer 1, it is important to recall that our paradigm did not impose temporal pressure on participants and that they were instructed to initiate movement with the mouse only after choosing the direction of the perceived movement. We have emphasized this crucial instruction in the revised methods (Participants were instructed to start moving only after reaching a decision, and were not given any particular instructions with regards to CoMs). We ensured that participants complied with this instruction, since the initial angle of the mouse trajectory corresponded in the vast majority of cases to the final response provided (90.36% of trials). This interpretation given by the reviewer, whereby participants move the mouse randomly and only then make a decision, is probably more compatible with experimental paradigms involving time pressure. Without time pressure, and according to our instructions, participants observed the stimulus until they thought they could discriminate its direction, and then triggered a motor command to click on the corresponding target, which in 90% of cases occurs along a curvilinear trajectory that does not include a change of direction. We refer to what happens after this initial movement as post-decisional, in line with the empirical literature and computational models that dichotomize the process into pre- and post-decisional parts. We agree with the reviewer that it would probably be better to consider that the decision-making process starts even before the participant moves the mouse and probably continues a few moments after the final response is given, but a complete overhaul in this direction would take us too far away from the framework that we adopted. We stressed this limitation in a new section of the discussion: “[...] we chose to dichotomize the response process into pre- and post-decisional stages depending on whether it occurred before or after the initial mouse movement. This dichotomy enabled us to interpret our modeling results within a powerful computational framework and rich literature. It also allowed for the operational definition of CoMs. However, it is also reductive since the decision-making process probably extended continuously along the mouse trajectory, leading to the final response.”

We have also reiterated in the revised methods that participants receive no instructions regarding changes of mind, these being defined a posteriori, and representing 9.64% of trials for which the initial angle of the trajectory does not correspond to the final response given (cf “Behavioural and neural markers of changes of mind” in the Result section of the article and the Behavioural analyses section in the original methods for the precise definition of CoMs).

An alternative interpretation is that the participant makes a first decision to start moving their mouse once they are reasonably confident in an answer (which quite sensibly occurs in preSMA), but then they continue to evaluate the stimulus and gather evidence the rest of the way (which seems to occur in the insula). This is similar to the authors’ existing story but does not require the strong assumption that people are continuing to decide after the decision has been made. As a side note, I think it is worth noting that the pre-SMA and insula both exhibit a substantial peak in their regression coefficients 0.4s “post decision”, suggesting that the pre-SMA is indeed still involved during that period. I would be wary of drawing strong conclusions about the difference between those pre-SMA and insula time courses based on Figure 4, especially given the sample size.

We fully agree with the reviewer’s interpretation. The only difference is that we consider the phase during which “they continue to evaluate the stimulus” an integral part of the decision-making process. We are not alone in this interpretation, which is common to all models of post-decisional evidence accumulation (Desender et al., 2021; van den Berg et al., 2016).

We also agree with the reviewer that, given the traces observed in Fig. 4, it is difficult to interpret the involvement of pSMA as purely decisional (i.e., stopping once the movement has been initiated). Based on neural data, the reviewer is right that we may have overinterpreted the differences observed between the pSMA and insula. A formal test of these differences would require evaluating a statistical interaction with ROI as a fixed effect. However, because sEEG electrodes are placed at different locations across patients, we do not have the statistical power to test this interaction with a between-subject design. We toned down our claim about differences between the pSMA and insula, and now mention the impossibility of testing for an interaction in the limitation section. We thank the reviewer for noting this important aspect we had overlooked.

Finally, our macro sEEG recordings did not allow us to assess the differences observed between the pSMA and insula statistically. Implementation schemes varied greatly across patients, and no patient had enough responsive channels in both regions to estimate

statistical interactions. Linking cortical activity to fine properties of evidence accumulation, such as the state of winning and losing accumulators will require single-unit recordings and multivariate analyses in future studies.

(3) I wonder about the authors' confidence model. Their model is that confidence is based on the amount of accumulated evidence 0 to 1s after the decision. Again, I question whether this activity is really occurring after the decision. If that were the case, it would imply that people would be unable to report their confidence when queried immediately after their decision. This does not seem plausible. Instead, if you think of that time as simply a slice of their decision time, then what you are measuring is the drift rate, which should indeed correlate with confidence. It would be useful to know whether this 1 s time window correlates more strongly with confidence than say the 0 to 1s window after stimulus onset. One would expect that to be true if the first few hundred milliseconds are noisy due to initial stimulus encoding, i.e. non-decision time. **But what about the interval from say 0.5s to 1.5s post stimulus onset? How does that compare?**

Our model uses the difference between accumulators to compute confidence. Therefore at the time of the decision, the winning accumulator is at a constant value corresponding to the decision bound, but the losing accumulator can be at any state between 0 and the bound. Therefore, within this modelling framework, participants could very well compute confidence immediately after the decision, based on the state of the losing accumulator (as in Kiani et al., 2014 for example). If we could measure the participant's confidence at each instant t however, we would observe that this confidence evolves with post-decisional time, as confirmed in previous studies (Desender et al., 2021; Yu et al., 2015). Our modelling results show that for most (87%) participants, adding post-decisional time improves model fits.

There is also a whole literature aimed at establishing the respective weight and role of model parameters over time (Desender et al., 2021; Herregods et al., 2023; Kiani et al., 2014; Moran et al., 2015; Moreno-Bote, 2010; Pleskac & Busemeyer, 2010), notably by quantifying the calibration of confidence depending on various decisional and postdecisional manipulations (Fleming et al., 2018). The drift rate is one of these parameters, but it is not the only one. Unfortunately, our paradigm is not optimal for assessing the role of drift rate as it was constant across trials. Indeed, sensory evidence was presented continuously to participants, with the stimulus remaining displayed on screen even after participants had moved the mouse, and we did not manipulate the difficulty (i.e. drift rate) due to time constraints in a clinical setting.

To attempt to answer the reviewer’s question about how confidence evolves over time, we tested how well simulated neural activity displayed in Figure 5 would relate to the accuracy of the response. We found that accuracy had an effect that increased over time, both when time-locking to stimulus onset (without considering non-decisional time) and when time-locking to movement onset. This result shows that to compute confidence ratings that optimally track the correctness of the upcoming decision, it is better to do it some time after the decision (corresponding to the movement onset in our conceptual framework) than in the 1 s following stimulus onset. Since the type of post-decisional model we used has received extensive empirical support for behavior, we will refrain from including this additional analysis in the manuscript and concentrate on better supporting the empirical validity of the model.

We opted for a readout occurring at a fixed latency after the decision (with some variability in the non-decision time), similar to previous works (Pereira et al., 2020; Pleskac & Busemeyer, 2010; van den Berg et al., 2016). We note that other studies have hypothesized that the readout occurs when accumulated evidence reaches additional collapsing bounds (Herregods et al., 2023; Moran et al., 2015), without significantly changing the signatures (Pleskac & Busemeyer, 2010; Sanders et al., 2016) of the so-computed confidence.

Figure. Regression coefficients (averaged across participants) for the effect of decision accuracy on simulated neural activity in Figure 5. Left: time-locked to the stimulus onset, Right: time-locked to the movement onset.

(4) Could the authors provide more concrete detail about the decision model? What does it mean that the accumulators are anti-correlated, and why this particular covariance factor of square root of 0.5?

The model assumes that noisy sensory information is accumulated over time by two accumulators, one per choice. There is one accumulator for the correct choice with a positive drift rate plus some noise and one accumulator for the incorrect choice with a negative drift rate plus some noise. The noise for each accumulator could be assumed to be independent ($\rho = 0$), but previous studies assume that it has negative cross-correlation, which makes the two accumulators inhibit each other (e.g. Kiani et al., 2014; Van den Berg et al., 2016). This is consistent with biological models of decision-making based on recurrent neural networks (Wang, 2002). Some fitting methods used in previous works (van den Berg et al., 2016) have only a limited choice in the values that this cross-correlation can take and chose $\rho = -\sqrt{0.5}$ to be close to the fitted values in other studies (Kiani et al., 2014).

In the present work, we used simulated log-likelihoods to fit the data, which allowed us to choose any cross-correlation coefficient. However, as we had limited time with the patients, we could not benefit from the thousands of trials that would be needed to have various levels of coherence and fit that value. Therefore, as we wrote in the methods, we decided to fix the value of ρ ($-\sqrt{0.5}$) to the value that was used by previous studies. A similar approach was used for the variability in the non-decision time, which was fixed to 60 ms.

We updated the corresponding methods section to better explain this choice and added some additional information about the model.

The noise for each accumulator could be assumed to be independent ($\rho = 0$) but previous studies assumed that it has negative cross-correlation which make the two accumulators inhibit each other (e.g. Kiani et al., 2014; Van den Berg et al., 2016). Therefore, to reduce the degrees of freedom of the model, we fixed the covariance between accumulators to $\rho = -0.5$, to match previous works (van den Berg et al., 2016). Similarly, we fixed the standard deviation of the non-decision time to 60 ms.

(5) Regarding exclusions - the authors exclude a lot of trials (17%). **Why give participants 6 s to decide but then exclude trials with decisions longer than 2.5 s?** Are the results robust to these exclusions? I also didn't understand the comment about how these longer decisions had "noisy trajectories", since the trajectories occur after the decision time. Why would they be related? If anything, I would think that trajectories would need to be more ballistic if a participant spent too long leaving the start point.

As mentioned above, we aimed to give participants as much time as possible to decide before moving the mouse. Our priority was to observe genuine CoMs that were not due to temporal pressure. However, we were confronted with clinical planning, and previous work confirmed that the 6-second delay left enough time for the participants while guaranteeing the possibility of collecting enough trials in the allotted time (Faivre et al., 2021).

Nevertheless, we are keen to test the robustness of our effects and have undertaken, under the reviewer's suggestion, to assess what happens if we include all trials in the analysis. At the level of individual channels, we identified 50 channels reflecting EA, with an 81% overlap with our original analysis. These results support our initial findings; we now mention them in the supplementary results. Concerning the ROI analyses, our results are broadly unchanged with regard to the links between RTs and HGA, although with slightly reduced effects in the insula and the IFG. They are virtually identical with regard to confidence and HGA. We included additional figures showing the results according to trial selection criteria at the end of this document. These figures are also presented in the revised supplementary results.

Additional analyses without exclusion criteria

To ensure that our exclusion criteria (excluding trials with erroneous responses or trials with overly delayed decision times) did not strongly impact our results we redid all of our analyses regarding the selection of evidence accumulation at the single channel level and the effects of decision time and confidence at the ROI level while including the previously excluded trials.

This manipulation left our results mostly unchanged, as illustrated in Fig. S8 for decision time and Fig. S9 for confidence.

Other more minor comments:

I didn't understand the point on p.7 where the authors state "Because of this low number of identified channels related to error processing, we focused the rest of the analysis on correct trials." Why should error trials be excluded? From the participant's perspective they don't know the difference between correct and error. Error trials are simply on the more difficult end of the spectrum than the correct trials. Why do the authors need to throw away a quarter of their data?

Our initial strategy was not to contaminate our confidence results with processes specific to error detection (Feuerriegel et al., 2022; Fu et al., 2019). We acknowledge that such processes were hardly observed in our paradigm, since we identified few channels showing a different level of HGA between correct and incorrect responses. In response to the

reviewer's request, we rerun all our analyses, including both correct and incorrect trials. We also transformed the confidence to be equivalent for errors and correct answers, taking the distance from 50%, which corresponds to $P(\text{correct}) = P(\text{incorrect})$. Thus, errors associated with low confidence (e.g. confidence = 20%, distance = 30) have a distance comparable to correct answers associated with high confidence (e.g. confidence = 80%, distance = 30). With these new analyses, we report overall similar results with regard to the effect of decision time and confidence on high-gamma activity, both in the pSMA and the Insula. The only major change was observed in the parietal cortex, where we observed a significant effect of decision time on high-gamma activity following stimulus onset, and an extended effect of confidence in the window leading to decision time (see comparative figure at the end of this document, which we also included in the revised SI).

Because the authors were using a staircase procedure to adjust the difficulty, do they have a trial-by-trial measure of the dot motion coherence? Could they use this variable to test whether the purported evidence accumulation regions had steeper slopes on the higher coherence trials, that participants were more confident on higher coherence trials, and that there were fewer changes of mind on higher coherence trials?

As indicated in response to a previous comment, the fact that we titrated coherence using a 1up/2down psychophysical staircase greatly limits the observed within-subject variance. Yet, we ran new analyses assessing the link between high gamma activity and decision times with sensory evidence (i.e., motion coherence) as a co-variate: $\text{HGA} \sim \text{decision time} * \text{sensory evidence}$. These new analyses performed at the level of regions of interest revealed no main effect of sensory evidence nor an interaction between decision times and sensory evidence (see below and revised SI). These analyses are not optimal as sensory evidence was titrated throughout the experiment using a 1up-2down staircase procedure, and therefore did not vary enough within participants to impact high gamma activity.

Figure. S7 Stimulus-aligned and movement-aligned high-gamma activity for regions of interest reflecting evidence accumulation (**A, B, C**) Top: Stimulus-aligned and movement-aligned high-gamma activity averaged across participants for all EA channels. Dark bars indicate a significant relationship between high-gamma activity and decision time ($p < 0.05$, FDR-corrected). Red bars indicate a significant relationship between high-gamma and motion coherence ($p < 0.05$, FDR-corrected) Blue lines correspond to averaged high-gamma for the 50% fastest trials and grey lines for the 50% slowest. Shaded areas correspond to 95% CI. Bottom: Regression coefficient as a function of time in the stimulus- and movement-aligned window. Light blue markers indicate the onset and offset times of significant effect, and the red marker indicates peak effects.

In a couple of places the authors try to argue that what they're seeing is not just a correlation with decision time. On page 9 they state that "we verified that the magnitude of the correlation between high-gamma activity and decision time was significantly lower than that of the correlations between high-gamma activity and confidence." I'm not sure what this is supposed to tell us. This difference in correlation could be due to the fact that decision time

is more noisy than the confidence measure, or because the relationship with confidence is more linear than with decision time. Later, they talk about interacting confidence and decision time. I did not understand the motivation behind this analysis or what it tells us about the relative contributions of confidence and decision time.

We thank the reviewer for pointing out this error on our part concerning the interpretation of the interaction between confidence and decision time. We replaced our original analysis with a new analysis relying on a full linear mixed-effects regression, including confidence and decision time as independent variables: ($H_{ga} \sim \text{Confidence} * \text{Decision Time} + (1 | \text{Participant/channel})$). This new analysis performed on the clusters of interest in the pSMA and the Insula shows a significant effect of confidence on high-gamma activity in the two ROIs, alongside significant effects of decision time for both ROIs in the move-aligned window. This new analysis confirms that high-gamma activity in the pSMA and Insula reflects confidence, even after the variance from decision times is accounted for. We have updated the manuscript accordingly.

Given some of the statements about OFC being involved in the output of the decision, I was surprised not to see any reference to the work on value based decision making, such as Camillo Padoa-Schioppa's recent work.

We thank the reviewer for this suggestion, which we now cite in the revised article when discussing the role of the OFC.

Ideally the mixed-effects regressions should include random slopes as well as intercepts to account for all sources of across-participant variability.

We agree that it would be ideal to include random slopes as well as intercepts to account for all sources of across-participant variability. However, as is often the case with these signals, full models failed to converge. We added this limitation in the revised methods: **Random slopes could not be added as they induced convergence failures.**

At one point in the paper the authors state that "These assumptions did not have much influence on our conclusions." This statement is too vague. To what extent did the authors explore different analyses/parameters?

It was indeed unclear, and we apologize for this. We now separate the assumptions on post-decisional evidence accumulation which do influence our conclusions and are now fully displayed in Fig. S6, and the assumption on the split of the non-decisional time which only affects the latency of the confidence effect. We meant that this later assumption had little

influence on our conclusions, considering the small range of plausible values (average non-decision times were short: average 156 ms \pm 28).

This part now reads:

To simulate neural data, we assumed that evidence accumulation would i) stop at the decision (Figure 5A, S6A), ii) continue post-decisionally up to the confidence readout (Figure S6B), or until the end of the trial(Figure 5B, S6C). We also assumed that the non-decision time would be split between a pre-accumulation non-decision time of 100 ms and the remaining non-decisional time would be a post-decision non-decision time corresponding to motor delays. The extent of this split slightly affected when confidence-related activity would begin prior to the movement onset.

Figure 1A - there is a reference to dark gray channels that were not selected for analysis but I do not see them in the figure.

We have corrected the legend of Figure 2.A.

Figure 3 - why isn't the green/gray distinction in the figure legend rather than in the caption? This was a strange departure from the other figures.

We have included this information in Figure 3.

Figure. S8 Results comparison between our original analysis (A), and similar analyses including error trials (B) and trials with decision times superior to 2.5s (C) Top (A to C): Template brain with channels reflecting evidence accumulation. The original channels are shown in blue, the channels specific to the analysis including errors in red, and the channels specific to the analysis including all decision times in purple. Bottom (A to C) Same convention as in Fig.4. Each row corresponds to a region of interest. Dark bars indicate a significant relationship between high-gamma activity and the decision time ($p < 0.05$, FDR-corrected). Blue lines correspond to averaged high-gamma activity for the 50% fastest trials and grey lines for the 50% slowest. Shaded areas correspond to 95% CI.

Figure. S9 Results comparison between our original analysis (A), and similar analyses including error trials (B) and trials with decision times superior to 2.5s (C) Top (A to C): Template brain with channels reflecting evidence accumulation. The original channels are shown in blue, the channels specific to the analysis including errors in red, and the channels specific to the analysis including all decision times in purple. Bottom (A to C) Same convention as in Fig.5. Each row corresponds to a region of interest. Dark bars indicate a significant relationship between high-gamma activity and confidence ($p < 0.05$, FDR-corrected). Green lines correspond to averaged high-gamma for the 50% trials with the highest confidence and grey lines for the 50% with the lowest confidence. Shaded areas correspond to 95% CI.

References

- Cecchi, R., Vinckier, F., Hammer, J., Marusic, P., Nica, A., Rheims, S., Trebuchon, A., Barbeau, E. J., Denuelle, M., Maillard, L., Minotti, L., Kahane, P., Pessiglione, M., & Bastin, J. (2022). Intracerebral mechanisms explaining the impact of incidental feedback on mood state and risky choice. *eLife*, 11, e72440. <https://doi.org/10.7554/eLife.72440>
- Desender, K., Donner, T. H., & Verguts, T. (2021). Dynamic expressions of confidence within an evidence accumulation framework. *Cognition*, 207, 104522. <https://doi.org/10.1016/j.cognition.2020.104522>
- Dou, W., Martinez Arango, L. J., Castaneda, O. G., Arellano, L., McIntyre, E., Yballa, C., &

- Samaha, J. (2023). *Neural Signatures of Evidence Accumulation Encode Subjective Perceptual Confidence* [Preprint]. Neuroscience.
<https://doi.org/10.1101/2023.04.28.538782>
- Faivre, N., Roger, M., Pereira, M., de Gardelle, V., Vergnaud, J.-C., Passerieux, C., & Roux, P. (2021). Confidence in visual motion discrimination is preserved in individuals with schizophrenia. *Journal of Psychiatry and Neuroscience*, 46(1), E65–E73.
<https://doi.org/10.1503/jpn.200022>
- Faivre, N., Vuillaume, L., Bernasconi, F., Salomon, R., Blanke, O., & Cleeremans, A. (2020). Sensorimotor conflicts alter metacognitive and action monitoring. *Cortex*, 124, 224–234. <https://doi.org/10.1016/j.cortex.2019.12.001>
- Feuerriegel, D., Murphy, M., Konksi, A., Mepani, V., Sun, J., Hester, R., & Bode, S. (2022). Electrophysiological correlates of confidence differ across correct and erroneous perceptual decisions. *NeuroImage*, 259, 119447.
<https://doi.org/10.1016/j.neuroimage.2022.119447>
- Fleming, S. M., van der Putten, E. J., & Daw, N. D. (2018). Neural mediators of changes of mind about perceptual decisions. *Nature Neuroscience*, 21(4), 617–624.
<https://doi.org/10.1038/s41593-018-0104-6>
- Fu, Z., Wu, D.-A. J., Ross, I., Chung, J. M., Mamelak, A. N., Adolphs, R., & Rutishauser, U. (2019). Single-Neuron Correlates of Error Monitoring and Post-Error Adjustments in Human Medial Frontal Cortex. *Neuron*, 101(1), 165-177.e5.
<https://doi.org/10.1016/j.neuron.2018.11.016>
- Gherman, S., Markowitz, N., Tostaeva, G., Espinal, E., Mehta, A. D., O'Connell, R. G., Kelly, S. P., & Bickel, S. (2023). *Intracranial electroencephalography reveals effector-independent evidence accumulation dynamics in multiple human brain regions* [Preprint]. Neuroscience. <https://doi.org/10.1101/2023.04.10.536314>
- Grogan, J. P., Rys, W., Kelly, S. P., & O'Connell, R. G. (2023). *Confidence is predicted by pre- and post-choice decision signal dynamics* [Preprint]. Neuroscience.
<https://doi.org/10.1101/2023.01.19.524702>

- Herregods, S., Denmat, P. L., Vermeylen, L., & Desender, K. (2023). *Modelling Speed-Accuracy Tradeoffs in the Stopping Rule for Confidence Judgments*.
<https://doi.org/10.1101/2023.02.27.530208>
- Kiani, R., Corthell, L., & Shadlen, M. N. (2014). Choice Certainty Is Informed by Both Evidence and Decision Time. *Neuron*, *84*(6), 1329–1342.
<https://doi.org/10.1016/j.neuron.2014.12.015>
- Le Denmat, P., Verguts, T., & Desender, K. (2024). A low-dimensional approximation of optimal confidence. *PLOS Computational Biology*, *20*(7), e1012273.
<https://doi.org/10.1371/journal.pcbi.1012273>
- Merrick, C. M., Dixon, T. C., Breska, A., Lin, J., Chang, E. F., King-Stephens, D., Laxer, K. D., Weber, P. B., Carmena, J., Thomas Knight, R., & Ivry, R. B. (2022). Left hemisphere dominance for bilateral kinematic encoding in the human brain. *eLife*, *11*, e69977. <https://doi.org/10.7554/eLife.69977>
- Moran, R., Teodorescu, A. R., & Usher, M. (2015). Post choice information integration as a causal determinant of confidence: Novel data and a computational account. *Cognitive Psychology*, *78*, 99–147. <https://doi.org/10.1016/j.cogpsych.2015.01.002>
- Moreno-Bote, R. (2010). Decision Confidence and Uncertainty in Diffusion Models with Partially Correlated Neuronal Integrators. *Neural Computation*, *22*(7), 1786–1811.
<https://doi.org/10.1162/neco.2010.12-08-930>
- O’Connell, R. G., Dockree, P. M., & Kelly, S. P. (2012). A supramodal accumulation-to-bound signal that determines perceptual decisions in humans. *Nature Neuroscience*, *15*(12), 1729–1735. <https://doi.org/10.1038/nn.3248>
- Pereira, M., Faivre, N., Iturrate, I., Wirthlin, M., Serafini, L., Martin, S., Desvachez, A., Blanke, O., Van De Ville, D., & Millán, J. del R. (2020). Disentangling the origins of confidence in speeded perceptual judgments through multimodal imaging. *Proceedings of the National Academy of Sciences*, *117*(15), 8382–8390.
<https://doi.org/10.1073/pnas.1918335117>
- Pereira, M., Megevand, P., Tan, M. X., Chang, W., Wang, S., Rezai, A., Seeck, M., Corniola,

- M., Momjian, S., Bernasconi, F., Blanke, O., & Faivre, N. (2021). Evidence accumulation relates to perceptual consciousness and monitoring. *Nature Communications*, *12*(1), 3261. <https://doi.org/10.1038/s41467-021-23540-y>
- Pleskac, T. J., & Busemeyer, J. R. (2010). Two-stage dynamic signal detection: A theory of choice, decision time, and confidence. *Psychological Review*, *117*(3), 864–901. <https://doi.org/10.1037/a0019737>
- Rausch, M., Zehetleitner, M., Steinhauser, M., & Maier, M. E. (2020). Cognitive modelling reveals distinct electrophysiological markers of decision confidence and error monitoring. *NeuroImage*, *218*, 116963. <https://doi.org/10.1016/j.neuroimage.2020.116963>
- Resulaj, A., Kiani, R., Wolpert, D. M., & Shadlen, M. N. (2009). Changes of mind in decision-making. *Nature*, *461*(7261), 263–266. <https://doi.org/10.1038/nature08275>
- Sanders, J. I., Hangya, B., & Kepecs, A. (2016). Signatures of a Statistical Computation in the Human Sense of Confidence. *Neuron*, *90*(3), 499–506. <https://doi.org/10.1016/j.neuron.2016.03.025>
- van den Berg, R., Anandalingam, K., Zylberberg, A., Kiani, R., Shadlen, M. N., & Wolpert, D. M. (2016). A common mechanism underlies changes of mind about decisions and confidence. *eLife*, *5*, e12192. <https://doi.org/10.7554/eLife.12192>
- Vickers, D. (1979). *Decision processes in visual perception*. Academic Press Inc.
- Wang, X.-J. (2002). Probabilistic decision making by slow reverberation in cortical circuits. *Neuron*, *36*(5), 955–968.
- Yu, S., Pleskac, T. J., & Zeigenfuse, M. D. (2015). Dynamics of postdecisional processing of confidence. *Journal of Experimental Psychology: General*, *144*(2), 489–510. <https://doi.org/10.1037/xge0000062>
- Zylberberg, A., Barttfeld, P., & Sigman, M. (2012). The construction of confidence in a perceptual decision. *Frontiers in Integrative Neuroscience*, *6*. <https://doi.org/10.3389/fnint.2012.00079>

Color code: The Reviewers' comments are in black, our responses in red, and existing or updated segments from the manuscript are highlighted in yellow

REVIEWER COMMENTS

Reviewer #1 (Remarks to the Author):

all my concerns have been addressed, congratulations to the authors

We appreciate the reviewer's feedback and thank them for their evaluation of our manuscript.

Reviewer #2 (Remarks to the Author):

I thank the authors for the clear and rigorous response to reviewer comments, and congratulate them on a very nice study. Although I still find the effect sizes to be modest (and brief in pSMA), I do appreciate the method-specific conventions and statistical issues, as well as the challenges involved in collecting this valuable dataset in a hospital setting. The added details and caveats about how to interpret EA-like signals in EEG data adequately address my concerns in the sense that it can be left up to the field to decide, and this paper provides a worthy addition to a growing and exciting literature.

We are happy to read that the reviewer is satisfied with the revised article. We thank them once again for their evaluation of our manuscript.

I have only one minor comment: wasn't sure about the header, "Response accuracy, decision times, and confidence *as* accumulated evidence" -- do you mean 'vs' or 'explained by' or 'as a function of'?

We meant 'explained by', this has been modified accordingly.

Reviewer #3 (Remarks to the Author):

The authors have addressed almost all of my concerns. However, they failed to adequately address my #1 concern, which was about the correlation between slope of the increase in

activity and decision time. I pointed out that slope is computed as the rise in activity divided by time, and so you are correlating (activity/time) ~ time. There is a mechanical correlation here because decision time is on both sides of the equation.

Consider the following possibility - activity rises to the same level with the same slope in every trial and stays there until the end of the trial, independent of the decision time. In trials with longer decision times, this would equate to a smaller slope than in trials with shorter decision times. As a result, you would find a negative correlation between slope and decision time, as the authors observe in those 41 channels.

I don't see how a permutation test addresses this issue. A permutation test simply tells us that when you shuffle the decision times on the right hand side of the correlation, the correlation goes away, as it would even if the correlation were mechanical.

This issue needs to be addressed more thoroughly/convincingly.

We thank the reviewer for clarifying their concern and apologize for not giving it due consideration. We have now dedicated time and effort to investigate this issue and found that it does not impact our main conclusions. In brief, we agree with the reviewer that there is a 'mechanical' correlation inherent in our analysis; however, if our results were solely based on this 'mechanical' correlation, we would obtain radically different results in two of our analyses (The ROI analysis presented in Fig. 4 and a new correlation analysis we now present in Fig. S14). In order to fully convince the reviewer (and ourselves) that it is the case, we conducted simulations with synthetic data. Before delving into the details of our response, we would like to note that the method for quantifying evidence accumulation in the brain is still debated (Frömer et al., 2024; O'Connell et al., 2024), especially when dealing with local field potential data, which prevents us from conducting analyses of variance that can be performed with single neuron data (Churchland et al., 2011).

In brief, our hypothesis posits that the signal ramps up from an initial value shortly after the stimulus onset to a boundary shortly before the response time. Let us call this the 'EA hypothesis'. The alternative hypothesis proposed by the reviewer suggests that *“activity rises to the same level with the same slope in every trial and stays there until the end of the trial,*

independent of the decision time". For simplification, we can call this the 'stim-locked hypothesis'.

To arbitrate between the two hypotheses, we simulated 10,000 response times resembling typical distributions, using a Gamma distribution plus 200 ms to avoid fitting slopes on too short windows (see the red trace on Figure R3_1). We then simulated 10,000 trials with a signal either linearly ramping until the response time (EA hypothesis), or ramping from 0 to 1 s then plateauing, independent of decision time, as expected under the stim-locked hypothesis. We added the same Gaussian noise to both simulations ($\mu = 0$, $\sigma = 0.1$). Note that the results of these simulations were independent of the parameters (amount of noise or plateau time).

Figure R3_1: heatmaps of simulated activity (sorted by decision time) for stim-locked (left) and EA (right) hypotheses. Dashed black lines represent the simulated decision time. The simulated decision time distribution over time is depicted in red. Purple dots show how neural activity at the decision time (*dec-t*) should be positively correlated with the decision-time under the stim-locked hypothesis but not under the EA hypothesis.

Confirmation of the soundness of the permutation analysis

We then computed the slope for each trial from stimulus onset to the decision time and correlated the resulting slope with decision time (as we did in the manuscript). In agreement with the reviewers' critique, we found correlations for both hypotheses, although at a much-reduced strength for the stim-locked hypothesis. We note that the overall strength of the correlations depended on the amount of noise in the simulation and cannot be compared to the data.

We repeated this correlation 1,000 times while permuting the decision time values across trials. The resulting distribution corresponded to the correlations obtained in the stim-locked hypothesis, suggesting that any spurious or mechanical correlation would also occur in the permutations. This goes against the reviewer's claim about our permutation procedure not being

efficient to compensate for this mechanical correlation. We conclude from this analysis that our procedure might be too liberal in selecting high-gamma activity instantiating EA, but that our permutation procedure controls for that and therefore, our results cannot be explained by spurious correlation.

Figure R3_2: distribution of permutated high-gamma activity slopes vs. decision time (in blue) compared to the correlation obtained with simulated data (green dot). No difference between the correlation coefficient (green) and the coefficient distribution after permuting (blue) is expected under the stim-locked hypothesis (left panel). In other words, we expect to identify a similar number of channels with observed and permutated data and, therefore no statistical significance, which is not what we found. Under the EA hypothesis (right panel), the permutated correlation coefficients are expected to be smaller than the ones obtained with the simulated data, leading to a number of EA-selective channels that is higher than what would have been obtained by chance, as we report in the manuscript. Furthermore, we computed the correlation between the level reached by neural activity at the decision time and the decision time (purple). This correlation should be positive under the stim-locked hypothesis (left) but null under the EA hypothesis (right). We found no empirical evidence in support of the predictions under the stim-locked hypothesis (see Fig. R3_4).

Confirmation of the soundness of the correlation with decision time over time.

The stim-locked hypothesis might explain the high-gamma slope vs. decision-time correlation found in *some channels* (we now clearly acknowledge this limitation; see below). However, if that were the case for all *channels*, first we would not find a significant number of *channels* with our permutation tests, and second, we would not find effects of decision times on stimulus-locked high-gamma activity. We illustrated this latter point by repeating our analysis on the simulated data under both hypotheses. This analysis confirms the absence of any difference

under the stim-locked hypothesis, while we observe a steeper ramp-up for short decision times under the EA hypothesis (Figure R3_3), consistent with Fig. 4 in the revised manuscript.

Figure R3_3: simulated activity time-locked to the stimulus for short and long decision times (dec-t). Time-resolved correlations with decision times are shown in green, with a negative correlation for the EA hypothesis (right).

This effect can be quantified by running a correlation between simulated activity at every time point and the decision time. Note that this analysis does not suffer from having decision times on both sides of the equation. We found no correlation under the stim-locked hypothesis and a negative correlation under the EA hypothesis, similar to what we found using hierarchical linear mixed-model analysis on high-gamma activity in Fig. 4 of the revised manuscript.

As a last control, we tested a direct prediction of the stim-locked hypothesis: high-gamma activity at the decision time should increase with increasing decision times. This should not occur under the EA hypothesis, as activity should coalesce to a similar level corresponding to the decision bound. Although we are careful not to draw strong conclusions from null results, we note that none of the channels selected through our initial procedure showed a significant correlation between high-gamma activity at the decision time and the decision-time itself (Spearman Rank test, one-sided, all p-values > 0.05, Fig. R3_4, Fig. S14), in contradiction with the stim-locked hypothesis' prediction.

Figure R3_4: Correlation between decision time and high-gamma activity at decision time. Each subplot corresponds to the correlation between high-gamma activity at decision time and decision time for a particular EA-candidate channel. Red background indicates the absence of a significant positive correlation.

We thank the reviewer for questioning our results and encouraging us to run these simulations, which now appear in the revised supplementary information. As we now clearly acknowledge in the revised manuscript, although the initial selection of EA *channels* might suffer from a bias due to this mechanical correlation, this bias is accounted for by our permutation test and could not lead to the correlations we found with our population analysis with linear mixed models. We now refer to EA-selective channels as EA-candidate channels, to emphasize that the link to EA is

also supported by the time-resolved negative correlation of high-gamma activity with decision times.

Finally, to ensure full disclosure of this mechanical correlation, we added all the analysis presented in the rebuttal to the SI, and we updated the limitation section as follows:

“Furthermore, we considered the link between the slope of neural activity and decision times as a functional marker of evidence accumulation. Although this is frequently used (Kiani et al., 2008; Mazurek et al., 2003; Ploran et al., 2007; Tremel & Wheeler, 2015), this marker remains debated. Indeed, one could argue that our definition of EA-candidate channels was somewhat circular, as it depended on the correlation between the slope of high-gamma activity and decision times within temporal windows that were themselves defined by decision times. To rule out this potential confound, we have performed additional simulations and analyses showing that our results were not accounted for by this circularity (see SI Fig. S12, S13 and Fig. S14). Furthermore, our analyses within ROIs, which assessed high-gamma activity as a function of decision time, are not impacted by this confound. Nevertheless, alternatives that consider sensory evidence in addition to decision times could be used in future experiments in which sensory evidence is not titrated to remain around the discrimination threshold.”

We thank the reviewer for raising this point, which pushed us to dig further into this issue and consolidate our trust in our results.

Frömer, R., Nassar, M. R., Etinger, B. V., & Shenhav, A. (2024). Common neural choice signals can emerge artefactually amid multiple distinct value signals. *Nature Human Behaviour*. <https://doi.org/10.1038/s41562-024-01971-z>

O’Connell, R. G., Corbett, E. A., Parés-Pujolràs, E., Feuerriegel, D., & Kelly, S. P. (2024). Regressing Away Common Neural Choice Signals does not make them Artifacts. Comment on Frömer et al (2024, Nature Human Behaviour). bioRxiv, 2024-09.

Gherman, S., Markowitz, N., Tostaeva, G., Espinal, E., Mehta, A. D., O’Connell, R. G., Kelly, S. P., & Bickel, S. (2024). Intracranial electroencephalography reveals

effector-independent evidence accumulation dynamics in multiple human brain regions. *Nature Human Behaviour*. <https://doi.org/10.1038/s41562-024-01824-9>

Churchland, Anne. K., Kiani, R., Chaudhuri, R., Wang, X.-J., Pouget, A., & Shadlen, M. N. (2011). Variance as a Signature of Neural Computations during Decision Making. *Neuron*, 69(4), 818-831. <https://doi.org/10.1016/j.neuron.2010.12.037>